# Task-Robust Model-Agnostic Meta-Learning

**Liam Collins**
ECE Department
University of Texas at Austin
Austin, TX 78712
liamc@utexas.edu

**Aryan Mokhtari**
ECE Department
University of Texas at Austin
Austin, TX 78712
mokhtari@austin.utexas.edu

**Sanjay Shakkottai**
ECE Department
University of Texas at Austin
Austin, TX 78712
sanjay.shakkottai@utexas.edu

## Abstract

Meta-learning methods have shown an impressive ability to train models that rapidly learn new tasks. However, these methods only aim to perform well in expectation over tasks coming from some particular distribution that is typically equivalent across meta-training and meta-testing, rather than considering worst-case task performance. In this work we introduce the notion of "task-robustness" by reformulating the popular Model-Agnostic Meta-Learning (MAML) objective [12] such that the goal is to minimize the maximum loss over the observed meta-training tasks. The solution to this novel formulation is task-robust in the sense that it places equal importance on even the most difficult and/or rare tasks. This also means that it performs well over all distributions of the observed tasks, making it robust to shifts in the task distribution between meta-training and meta-testing. We present an algorithm to solve the proposed min-max problem, and show that it converges to an $\epsilon$-accurate point at the optimal rate of $\mathcal{O}(1/\epsilon^2)$ in the convex setting and to an $(\epsilon, \delta)$-stationary point at the rate of $\mathcal{O}(\max\{1/\epsilon^5, 1/\delta^5\})$ in nonconvex settings. We also provide an upper bound on the new task generalization error that captures the advantage of minimizing the worst-case task loss, and demonstrate this advantage in sinusoid regression and image classification experiments.

## 1 Introduction

Despite continual advances in computational power and data collection, many scenarios remain in which machine learning models must rapidly adapt to previously unseen tasks. Motivated by such scenarios, meta-learning techniques aim to learn how to learn quickly from few samples by leveraging knowledge acquired while learning prior tasks [4, 35]. The recent successes of these techniques in areas such as few-shot learning [12, 31, 33, 37] and reinforcement learning [8, 34, 38] have sparked tremendous interest in meta-learning.

Following the setting introduced in [3], most offline meta-learning methods try to minimize the expected loss on new tasks drawn from the same, but unknown, distribution as a finite set of meta-training tasks. For example, in gradient-based meta-learning, the learning method is typically a small number of stochastic gradient descent (SGD) steps, and the means to learn quickly is having a favorable initialization. Standard methods thus try to find an initialization that enables the model fine-tuned via task-specific SGD to perform well in expectation over new tasks. Since they assume

the new tasks are drawn from the same unknown distribution as the meta-training tasks, during meta-training they attempt to minimize the average empirical loss after one step of SGD [12, 26].

However, by minimizing the average loss, such methods may perform arbitrarily poorly on difficult and/or rare meta-training tasks. In many cases, a model that performs well across all tasks is desired, even the most difficult and rare tasks. Consider for example applications in which safety is critical, such as object detection in self-driving cars, in which failing to detect rarely seen objects may result in driving accidents. In this and similar settings, the failure of the system to produce accurate results for the worst-case task could possibly cause severe issues. Moreover, existing methods' disregard for worst-case performance relies on the often unrealistic assumption that the meta-test tasks are drawn from the same distribution as the meta-training tasks. If the meta-training dataset overestimates the prevalence of certain types of tasks in the meta-test distribution, existing methods will overfit to the popular tasks and fail to generalize to new tasks in both expectation and in the worst case. Indeed, existing generalization bounds for gradient-based meta-learning strategies depend on the similarity of the meta-test tasks to the meta-training solution [41, 2], rather than exploiting the diversity of the meta-training tasks to show generalization to a broad range of new tasks. To address these issues, we propose a novel meta-learning formulation that calls for minimizing the maximum as opposed to average task loss during meta-training. Our contributions are threefold:

- We modify the standard gradient-based meta-learning framework, Model-Agnostic Meta-Learning (MAML) [12], to find an initialization that minimizes the loss after one SGD step for the *worst-case* task, where tasks are broadly defined as distributions over few-shot learning problems. Our new formulation, Task-Robust MAML (TR-MAML), thus yields a "task-robust" solution, in the sense that it prioritizes performance equally on all observed tasks, including the hardest and rarest ones. Importantly, this means it is also robust to all shifts in distribution over the sampled tasks from meta-training to meta-testing.

- We present an algorithm to solve our min-max formulation and prove that it convergences efficiently in both convex and nonconvex settings. In the convex case, it achieves the optimal rate of $\mathcal{O}(\epsilon^{-2})$ stochastic gradient evaluations, and in the nonconvex case, it reaches an $(\epsilon, \delta)$-stationary point at a rate of $\mathcal{O}(\max\{\epsilon^{-5}, \delta^{-5}\})$ stochastic gradient evaluations.

- We capture the generality of our formulation's task robustness by giving a Rademacher complexity bound on the generalization error of any new task within the convex hull of the meta-training tasks, as well as showing improved performance in few-shot sinusoid regression and image classification experiments compared to MAML.

**Related Work.** Among a variety of meta-learning formulations, MAML [12] has become especially popular due to its efficiency and flexibility, inspiring many follow-up works [1, 22, 26, 5, 21]. From more theoretical perspectives, [11] analyzed the convergence of MAML with nonconvex losses, [30] and [41] presented MAML variants with guarantees in both convex and nonconvex settings, and other works have shown regret bounds for online analogues of MAML [13, 42, 18]. Meanwhile, robustness in meta-learning has been studied in multiple recent works. In [43] and [40], the authors proposed models whose expected performance is robust to perturbations in the task samples, and [20] extended MAML to deal with imbalances in the number of samples per task instance and out-of-distribution meta-test tasks, but their model requires a complicated dataset encoding and computing per-task balancing variables. Additionally, In [15] a heuristic was introduced that aims to prevent over-performing on certain meta-training tasks by regularizing the inequality among task losses, although only across mini-batches. [6] also considered a task-weighted objective and showed Rademacher complexity-based generalization bounds, but their weights utilize task similarity to a particular target rather than optimizing for worst-case performance. To the best of our knowledge, no other offline meta-learning formulation attempts to minimize the worst-case loss over tasks.

Many works outside meta-learning have considered min-max optimization problems of the finite-sum form discussed here. In the context of distributionally-robust optimization, [32] and [10] argued that minimizing the maximal loss over a set of possible distributions can provide better generalization performance than minimizing the average loss. While the stochastic mirror descent-ascent algorithm achieves the asymptotically optimal $\mathcal{O}(\epsilon^{-2})$ convergence rate to an $\epsilon$-accurate solution in the convex setting [25], the literature is less established for nonconvex problems. In [29], the authors proposed a stochastic inexact proximal point method that attains $\tilde{\mathcal{O}}(\epsilon^{-6})$ convergence in terms of the outer minimization problem when that problem is nonsmooth and weakly convex, while in [28] $\tilde{\mathcal{O}}(\epsilon^{-4})$ convergence was shown when the outer problem is smooth and strongly convex. In the

deterministic case, the authors of [27] demonstrated an $\tilde{\mathcal{O}}(\epsilon^{-3.5})$ convergence rate to an $\epsilon$-first-order Nash equilibrium for a gradient descent-ascent algorithm. Also, [7] and [16] analyzed first-order methods that improve on these rates but rely on an oracle to solve the inner maximization.

## 2 Problem Formulation

Before discussing our min-max objective, we first formalize the meta-learning scenario. Let $x \in \mathcal{X}$ and $y \in \mathcal{Y}$ denote inputs and labels, respectively, and let $h_w : \mathcal{X} \rightarrow \mathcal{Y}$ represent the model parameterized by $w$. The performance of $h_w$ on a point $(x, y) \in \mathcal{X} \times \mathcal{Y}$ is determined by $\ell(h_w(x), y)$, where $\ell : \mathcal{Y} \times \mathcal{Y} \rightarrow \mathbb{R}_+$ is a loss function, e.g., the mean squared error in regression and the cross entropy loss in classification. We define a task $\mathcal{T}_i$ as a distribution $\mathcal{D}_i$ over task instances, which are few-shot learning episodes composed of two data batches, $D_{i,j}^{\text{train}}$ and $D_{i,j}^{\text{test}}$, of $K$ and $J$ points, respectively, in $\mathcal{X} \times \mathcal{Y}$. Within each task instance, the goal of the learner is to perform well on the points in $D_{i,j}^{\text{test}}$ after learning from the points in $D_{i,j}^{\text{train}}$, which is made possible by assuming that each point in both batches is an i.i.d. sample from the same distribution $\mathcal{D}_{i,j}$ over $\mathcal{X} \times \mathcal{Y}$.

During meta-training, a finite number of task instances are observed by first sampling a task $\mathcal{T}_i$ from $P(\mathcal{T})$, the meta-training distribution over tasks, then sampling $(D_{i,j}^{\text{train}}, D_{i,j}^{\text{test}}) \sim \mathcal{D}_i$. Let there be $m_i$ instances of the $i$-th task for each of $n$ tasks observed during meta-training, for a total of $m := \sum_{i=1}^n m_i$ task instances. In MAML, for each task instance, the dataset $D_{i,j}^{\text{train}}$ is used to update a global initialization $w$ via one SGD step with respect to the expected loss of the model on $\mathcal{D}_{i,j}$, namely $f_{i,j}(w) := \mathbb{E}_{(x,y) \sim \mathcal{D}_{i,j}}[\ell(h_w(x), y)]$. Afterwards, the resulting "test" loss is approximated using $D_{i,j}^{\text{test}}$, which serves as the meta-training loss. With the ultimate goal of learning how to learn new task instances coming from the same distribution $P(\mathcal{T})$, the meta-training objective is to find a $w$ that minimizes the post-update loss on $D_{i,j}^{\text{test}}$ on average over the observed task instances, namely:

$$\min_{w \in \mathcal{W}} \frac{1}{m} \sum_{i=1}^n \sum_{j=1}^{m_i} \hat{f}_{i,j}(w - \alpha \nabla \hat{f}_{i,j}(w; D_{i,j}^{\text{train}}), D_{i,j}^{\text{test}}), \tag{1}$$

where $\alpha$ is the inner update step size and $\hat{f}_{i,j}(\cdot, D_{i,j}^{\text{test}}) = \frac{1}{J} \sum_{(x,y) \in D_{i,j}^{\text{test}}} \ell(h_w(x), y)$ is the sample-average approximation of $f_{i,j}(\cdot)$ using the $J$ samples in $D_{i,j}^{\text{test}}$, and likewise for $\nabla \hat{f}_{i,j}(\cdot, D_{i,j}^{\text{train}})$. As referred to in the introduction, the solution of (1) may perform arbitrarily poorly on tasks that differ significantly from the average task instance, which is especially problematic if tasks similar to those become more prevalent at meta-test time due to a distributional shift. Thus, we propose to treat all $n$ meta-training tasks equally by minimizing the maximum task empirical average meta-loss $\hat{F}_i(w)$:

$$\min_{w \in \mathcal{W}} \max_{i \in [n]} \left\{ \hat{F}_i(w) := \frac{1}{m_i} \sum_{j=1}^{m_i} \hat{f}_{i,j}(w - \alpha \nabla \hat{f}_{i,j}(w, D_{i,j}^{\text{train}}), D_{i,j}^{\text{test}}) \right\}. \tag{2}$$

Problem (2) is equivalent to the problem of finding the $w^*$ that minimizes the worst-case meta-learning performance over all distributions of the $n$ tasks, since the worst-case distributions will occur at the extreme points of the probability simplex in $n$ dimensions. We write this relaxed problem as

$$\min_{w \in \mathcal{W}} \max_{p \in \Delta_n} \left\{ \phi(w, p) := \sum_{i=1}^n p_i \hat{F}_i(w) \right\}, \tag{3}$$

where $p_i$ is the probability associated with task $i$, the vector $p = (p_1, \ldots, p_n)$ is the concatenation of probabilities, and $\Delta_n = \{p \in \mathbb{R}_+^n \mid \sum_{i=1}^n p_i = 1\}$. Note that (3) may be hard to solve if $n$ is very large, and in many applications, $m$ is indeed very large. However, $n$ need not be, as tasks may be defined to encompass many similar task instances. We provide experiments for this case in Section 6.

By optimizing for worst-case performance, the formulation in (3) encourages a solution $w^*$ that performs similarly across all of the observed tasks. Instead of disregarding performance on some tasks, any algorithm that solves (3) must try to perform reasonably well on all of them. Indeed, as observed in [9], the min-max formulation implicitly regularizes the variance of the losses. This naturally makes the solution robust to distributional shifts between meta-training and meta-testing, and we provably show its ability to generalize to new tasks in Section 5.

## 3 Algorithm

Taking inspiration from [25], we propose to solve the meta-training problem (3) using a Euclidean version of the robust stochastic mirror-prox algorithm. Our method, termed TR-MAML and outlined in Algorithm 1, requires stochastic gradient estimates of the function $\phi(w, p)$ defined in (3) with respect to $w$ and $p$. Note that the full gradients, denoted by $g_w(w, p)$ and $g_p(w, p)$, respectively, are

$$g_w(w, p) = \sum_{i=1}^{n} \frac{p_i}{m_i} \sum_{j=1}^{m_i} (I - \alpha \nabla^2 \hat{f}_{i,j}(w, D_{i,j}^{\text{train}})) \nabla \hat{f}_{i,j}(w - \alpha \nabla \hat{f}_{i,j}(w, D_{i,j}^{\text{train}}), D_{i,j}^{\text{test}}), \quad (4)$$

$$g_p(w, p) = \left[ \frac{1}{m_i} \sum_{j=1}^{m_i} \hat{f}_{i,j}(w - \alpha \nabla \hat{f}_{i,j}(w, D_{i,j}^{\text{train}}), D_{i,j}^{\text{test}}) \right]_{1 \leq i \leq n}, \quad (5)$$

where $\nabla^2 \hat{f}_{i,j}(w, D_{i,j}^{\text{train}})$ is the sample average approximation of $\nabla^2 f_{i,j}(w)$ based on the $K$ samples in $D_{i,j}^{\text{train}}$, and the notation $[a_i]_{1 \leq i \leq n}$ corresponds to the vector $[a_1, \ldots, a_n] \in \mathbb{R}^n$. Since $n$ and the $m_i$'s may be large, TR-MAML must estimate the full gradients $g_w$ and $g_p$ on each iteration. To do so, it first uniformly and independently samples a set $\mathcal{C}$ of $C$ indices $\{i_k\}_{k=1}^{C}$ from $\{1, \ldots, n\}$. For each $i_k \in \mathcal{C}$, the algorithm samples one index $j_k$ uniformly from $\{1, \ldots, m_i\}$, then estimates $g_w(w, p)$ and $g_p(w, p)$ using the data $\{(D_{i_k,j_k}^{\text{train}}, D_{i_k,j_k}^{\text{test}})\}_{k=1}^{C}$. The two estimates can then be written as

$$\hat{g}_w(w, p) = \frac{n}{C} \sum_{k=1}^{C} p_{i_k} (I - \alpha \nabla^2 \hat{f}_{i_k,j_k}(w, D_{i_k,j_k}^{\text{train}})) \nabla \hat{f}_{i_k,j_k}(w - \alpha \nabla \hat{f}_{i_k,j_k}(w, D_{i_k,j_k}^{\text{train}}), D_{i_k,j_k}^{\text{test}}), \quad (6)$$

$$\hat{g}_p(w, p) = \frac{n}{C} \sum_{k=1}^{C} \hat{f}_{i_k,j_k}(w - \alpha \nabla \hat{f}_{i_k,j_k}(w, D_{i_k,j_k}^{\text{train}}), D_{i_k,j_k}^{\text{test}}) e_{i_k}, \quad (7)$$

where $e_{i_k}$ is the $i_k$-th standard basis vector in $\mathbb{R}^n$. We show that $\hat{g}_w(w, p)$ and $\hat{g}_p(w, p)$ are unbiased and bound their second moments in Section 4. In order to solve (3), TR-MAML initializes $p^0 = [1/n]_{1 \leq i \leq n}$ and $w^0 \in \mathcal{W}$, then executes alternating projected stochastic gradient descent-ascent. In particular, from iterations $t = 0$ to $T - 1$, TR-MAML computes $w^{t+1}$ and $p^{t+1}$ as

$$w^{t+1} = \Pi_{\mathcal{W}}(w^t - \eta_w^t \hat{g}_w(w^t, p^t)), \quad p^{t+1} = \Pi_{\Delta_n}(p^t + \eta_p^t \hat{g}_p(w^t, p^t)), \quad (8)$$

where $\eta_w^t$ and $\eta_p^t$ are step sizes, $\Pi_W(u) = \arg\min_{w \in \mathcal{W}} \|u - w\|_2$ and $\Pi_{\Delta_n}(q) = \arg\min_{p \in \Delta_n} \|p - q\|_2$. The projections are convex programs and can be solved efficiently using standard techniques. In particular, since $\Delta_n$ is the full simplex, $\Pi_{\Delta_n}(q)$ can be computed in $\mathcal{O}(n \log n)$ time [39]. As mentioned previously, tasks can be defined to leverage similarity among the task instances such that $n$ is small, in which case the $\mathcal{O}(d^2)$ per-iteration cost of both MAML and TR-MAML due to the Hessian estimations trivializes the added cost of the simplex projection in TR-MAML, thus TR-MAML has effectively the same computational cost as MAML. Nevertheless, first-order MAML approximations [12, 26, 11] may be seamlessly applied to TR-MAML to reduce the computational burden. After $T$ iterations, TR-MAML terminates in one of two ways: **Case T1.** If each $\hat{F}_i(w)$ is convex, TR-MAML outputs $w_T^c := \frac{1}{T} \sum_{t=1}^{T} w^t$ and $p_T^c := \frac{1}{T} \sum_{t=1}^{T} p^t$. **Case T2.** Otherwise, TR-MAML samples $\tau$ uniformly from $\{1, ..., T\}$ and outputs $w_T^\tau := w^\tau$ and $p_T^\tau := p^\tau$.

## 4 Convergence Analysis

We next analyze the convergence of TR-MAML to a solution of (3). Convergence results for stochastic gradient-based algorithms typically assume access to unbiased stochastic gradients with bounded second moments [25, 29]. In our case, $\hat{g}_w$ and $\hat{g}_p$ are naturally unbiased, but bounding their second moments requires modest assumptions on the functions $\hat{f}_{i,j}$ due to the nested structure of $\hat{F}_i$.

**Assumption 1.** $\hat{f}_{i,j}(\cdot, D_{i,j}^{train})$ and $\hat{f}_{i,j}(\cdot, D_{i,j}^{test})$, $\forall i \in [n]$ and $j \in [m_i]$ are $\hat{B}$-bounded and $\hat{L}$-Lipschitz. Furthermore, $\lambda_{\min}(\nabla^2 \hat{f}_{i,j}(w, D_{i,j}^{train})) \geq -\hat{H}$ for all $w \in \mathcal{W}$.

With this assumption, we can bound the second moments. All proofs are given in the appendix.

**Lemma 1.** Under Assumption 1, for all $w \in \mathcal{W}, p \in \Delta_n$, vectors $\hat{g}_w(w, p)$ and $\hat{g}_p(w, p)$ satisfy: (i) $\mathbb{E}[\hat{g}_w(w, p)] = g_w(w, p), \mathbb{E}[\hat{g}_p(w, p)] = g_p(w, p)$; and (ii) Bounded second moment: $\mathbb{E}[\|\hat{g}_w\|_2^2] \leq n(1 + \alpha \hat{H})^2 \hat{L}^2$; $\mathbb{E}[\|\hat{g}_p\|_2^2] \leq \frac{n(n+C+1)\hat{B}^2}{C} =: \hat{G}_p^2$

**Algorithm 1** Task-Robust MAML (TR-MAML)

---

**Input:** $m$ task instances of $n$ unique tasks; parameters $\alpha$, $\{\eta_w^t\}_t$, $\{\eta_p^t\}_t$, $T$, $C$

Initialize $p^1 = [\frac{1}{n}]_{1 \le i \le n}$ and $w^1 \in \mathcal{W}$ arbitrarily.

**for** $t = 0$ **to** $T - 1$ **do**

    Sample a batch $\mathcal{C}$ of $C$ unique task indices uniformly from $\{1, \ldots, n\}$.

    **for** $i_k \in \mathcal{C}$ **do**

        Sample one task instance index $j_k$ uniformly from $\{1, \ldots, m_{i_k}\}$.

    **end for**

    Compute $\hat{g}_w(w^t, p^t)$ and $\hat{g}_p(w^t, p^t)$ using (6) and (7), respectively.

    Update $w^{t+1}$ and $p^{t+1}$ as in (8).

**end for**

**Output:** See Cases T1 and T2.

---

**Convex Setting.** Our first convergence result holds in the case when each $\hat{F}_i$ is convex. Note that the convexity of each $f_{i,j}$ does not imply the convexity of $\hat{F}_i$ (consider as a counterexample $f_{i,j}(w) = 1/w$ for $w \in \mathbb{R}_+ \setminus \{0\}$). In Lemma 2 we adapt a result from [13] showing that the strong convexity of each $\hat{f}_{i,j}(\cdot, D_{i,j}^{\text{test}})$ implies the strong convexity of $\hat{F}_i$ under an additional assumption on each $\hat{f}_{i,j}(\cdot, D_{i,j}^{\text{train}})$.

**Assumption 2.** $\hat{f}_{i,j}(\cdot, D_{i,j}^{train})$, for all $j \in [m_i]$, is $\hat{M}$-smooth and $\hat{\rho}$-Hessian-Lipschitz.

**Lemma 2.** *(Adapted from [13], Theorem 1) Suppose $\alpha < 1/\hat{M}$ and Assumptions 1-2 hold. If $\hat{f}_{i,j}(\cdot, D_{i,j}^{test})$ is $\hat{\mu}$-strongly convex $\forall j \in [m_i]$, then $\hat{F}_i$ is $\tilde{\mu} := (\hat{\mu}(1 - \alpha\hat{M})^2 - \alpha\hat{L}\hat{\rho})$-strongly convex.*

The optimal rate of convergence for solving convex-concave stochastic min-max problems is $\mathcal{O}(1/\epsilon^2)$, where convergence rate is measured in terms of the expected number of stochastic gradient computations required to achieve a duality gap of $\epsilon$ [25]. The duality gap of the pair $(\tilde{w}, \tilde{p})$ is defined as $\max_{p \in \Delta_n} \phi(\tilde{w}, p) - \min_{w \in \mathcal{W}} \phi(w, \tilde{p})$. By strong duality, $(\tilde{w}, \tilde{p})$ is optimal if and only if it has a duality gap of zero. We show that TR-MAML achieves the optimal $\mathcal{O}(1/\epsilon^2)$ rate by adapting Theorem 2 from [24], which in turn is a simplified version of Theorem 1 from [17].

**Theorem 1.** *(Adapted from [24], Theorem 2) Consider problem (3) when each $\hat{F}_i$ is convex and Assumption 1 holds. Suppose there exists a ball of radius $R_{\mathcal{W}}$ that contains $\mathcal{W}$. With step sizes $\eta_w = 2R_{\mathcal{W}}/((1 + \alpha\hat{H})\hat{L}\sqrt{nT})$ and $\eta_p = 2/(\hat{G}_p\sqrt{T})$, the output of TR-MAML satisfies:*

$$\mathbb{E}\left[\max_{p \in \Delta_n} \phi(w_T^{\boldsymbol{c}}, p) - \min_{w \in \mathcal{W}} \phi(w, p_T^{\boldsymbol{c}})\right] \le \frac{3\sqrt{n}R_{\mathcal{W}}(1 + \alpha\hat{H})\hat{L} + 3\hat{G}_p}{\sqrt{T}}$$

Thus, TR-MAML requires $T = \mathcal{O}(1/\epsilon^2)$ iterations to reach an expected duality gap of at most $\epsilon$. Since it computes a constant number of stochastic oracle evaluations per iteration, its convergence rate is the optimal $\mathcal{O}(1/\epsilon^2)$ stochastic oracle calls to reach an $\epsilon$-accurate solution.

**Nonconvex Setting.** We next study the case when each $\hat{F}_i$ may be nonconvex and as a result, $\phi(w, p)$ may be nonconvex in $w$. Here we must evaluate the pair $(w_T^\tau, p_T^\tau)$ returned by our algorithm differently with respect to $p$ and $w$: we still intend that $p_T^\tau \in \Delta_n$ globally maximizes $\phi(w_T^\tau, \cdot)$, but can only hope to find $w_T^\tau$ near a stationary point of $\phi(\cdot, p_T^\tau)$. Thus, we say that $(\tilde{w}, \tilde{p})$ is an $(\epsilon, \delta)$-stationary point of $\phi$ if

$$\|\nabla_w \phi(\tilde{w}, \tilde{p})\|_2 \le \epsilon \quad \text{and} \quad \phi(\tilde{w}, \tilde{p}) \ge \max_{p \in \Delta_n} \phi(\tilde{w}, p) - \delta, \tag{9}$$

where $\epsilon, \delta > 0$, assuming that $\mathcal{W} = \mathbb{R}^d$, otherwise we consider the projected gradient, which we discuss later. In either case we will leverage smoothness. The function that we aim to minimize, $\max_{p \in \Delta_n} \phi(w, p)$, is non-smooth because of the maximization, but we can again adapt a result from [13] to show that each $\hat{F}_i$ is smooth under the previous assumptions on each $\hat{f}_{i,j}$.

**Lemma 3.** *(Adapted from [13], Theorem 1) Under Assumptions 1 and 2, each $\hat{F}_i$ is $\tilde{M}$-smooth, where $\tilde{M} := \hat{M}(1 + \alpha\hat{M})^2 + \alpha\hat{L}\hat{\rho}$.*

We must also compute the expected squared deviation of the stochastic gradient $\hat{g}_w$, denoted by $\sigma_w^2$.

**Lemma 4.** *For all $w \in \mathcal{W}$ and $p \in \Delta_n$,*

$$\sigma_w^2(w,p) := \mathbb{E}[\|\hat{g}_w(w,p) - g_w(w,p)\|_2^2] = \frac{n}{C}\sigma^2(w,p) + \frac{n}{C}\sum_{i=1}^n \sigma_i^2(w,p) \qquad (10)$$

*where $\sigma^2(w,p) := \sum_{i=1}^n \|p_i \nabla \hat{F}_i(w) - \frac{1}{n}\sum_{i'=1}^n p_{i'} \nabla \hat{F}_{i'}(w)\|_2^2$ and*
$\sigma_i^2(w,p) := \frac{p_i^2}{m_i}\sum_{j=1}^{m_i} \|(I - \alpha \nabla^2 \hat{f}_{i,j}(w, D_{i,j}^{train}))\nabla \hat{f}_{i,j}(w - \alpha \nabla^2 \hat{f}_{i,j}(w, D_{i,j}^{train}), D_{i,j}^{test}) - \nabla \hat{F}_i(w)\|_2^2$.

Here $\sigma^2$ represents the inter-task variance and each $\sigma_i^2$ represents an intra-task variance. With $\sigma_w^2$ defined, the convergence of TR-MAML when $\mathcal{W} = \mathbb{R}^d$ can be shown via the following theorem.

**Theorem 2.** *If Assumptions 1 and 2 hold, $\mathcal{W} = \mathbb{R}^d$ and $\eta_w^t = T^{-\beta}$, and $\eta_p^t = \sqrt{2}(T^{2\beta}\hat{G}_p)^{-1}$ for all $t = 1, \ldots, T$ and any $\beta \in (0, \frac{1}{2})$, and $T^\beta > \tilde{M}/2$, then the output of Algorithm 1 satisfies*

$$\mathbb{E}\left[\|\nabla_w \phi(w_T^\tau, p_T^\tau)\|_2^2\right] \le \frac{\phi(w^1, p^1) + \hat{B} + \sqrt{2n}\hat{B} + 2\tilde{M}\sigma_w^2}{T^\beta - \tilde{M}/2},$$

$$\mathbb{E}\left[\phi(w_T^\tau, p_T^\tau)\right] \ge \max_{p \in \Delta_n}\{\mathbb{E}\left[\phi(w_T^\tau, p)\right]\} - \hat{G}_p/(\sqrt{2}T^{\min\{2\beta, 1-2\beta\}}).$$

Theorem 2 shows that Algorithm 1 converges in expectation to an $(\epsilon, \delta)$-stationary point of $\phi$ in $\mathcal{O}(\max\{1/\epsilon^{2/\beta}, 1/\delta^{1/\min\{2\beta, 1-2\beta\}}\})$ stochastic gradient evaluations in the unconstrained setting. Note that $\beta$ can be tuned to favor convergence with respect to $w$ or $p$. To treat convergence with respect to $w$ and $p$ equally, the optimal setting is $\beta = \frac{2}{5}$, yielding a convergence rate of $\mathcal{O}(\max\{1/\epsilon^5, 1/\delta^5\})$.

We finally consider the case when $\mathcal{W}$ is a compact, convex set. In this setting the notion of an $(\epsilon, \delta)$-stationary point must be altered such that $\epsilon$ upper bounds the projected gradient, $\bar{g}_w$, defined as

$$\bar{g}_w(w^t, p^t) := \frac{1}{\eta_w^t}(w^t - \Pi_{\mathcal{W}}(w^t - \eta_w^t \hat{g}_w(w^t, p^t))),$$

since this vector reveals how much the solution can be improved by moving within the feasible set. In the following theorem, we choose $C$ as a function of $T$ to show convergence.

**Theorem 3.** *Suppose Assumptions 1 and 2 hold. Let $\tilde{\sigma}_w^2 := C\sigma_w^2$ and set $\eta_w^t = 1/(2\tilde{M})$ and $\eta_p^t = (T^\beta \hat{B}\sqrt{n})^{-1}$ for $t \in [T]$, and the task batch size as $C = T^\beta$, for any $\beta \in (0, 1)$, then*

$$\mathbb{E}\left[\|\bar{g}_w(w_T^\tau, p_T^\tau)\|_2^2\right] \le \frac{8\tilde{M}(\phi(w^1, p^1) + \hat{B})}{3T} + \frac{8\tilde{M}\hat{B}\sqrt{n} + 4\tilde{\sigma}_w^2}{3T^\beta},$$

$$\mathbb{E}\left[\phi(w_T^\tau, p_T^\tau)\right] \ge \max_{p \in \Delta_n}\{\mathbb{E}\left[\phi(w_T^\tau, p)\right]\} - \frac{\hat{B}\sqrt{n}}{T^{\min\{\beta, 1-\beta\}}}.$$

The number of stochastic gradient evaluations is now $\mathcal{O}(CT) = \mathcal{O}(T^{1+\beta})$, so Theorem 3 shows Algorithm 1 converges to an $(\epsilon, \delta)$-stationary point after $\mathcal{O}(\max\{1/\epsilon^{(2+2\beta)/\beta}, 1/\delta^{(1+\beta)/\min\{\beta, 1-\beta\}}\})$ evaluations with convex, compact $\mathcal{W}$ and nonconvex $\hat{F}_i$. By setting $\beta = \frac{2}{3}$ we treat convergence with respect to $w$ and $p$ equally, yielding a complexity of $\mathcal{O}(\max\{1/\epsilon^5, 1/\delta^5\})$ evaluations.

## 5 Generalization Bounds

Given that the meta-learner has access to a finite number of task instances during meta-training, there are two types of generalization to consider: generalization to new instances of previously-seen tasks, and generalization to new tasks. We start by bounding the error on new instances of previously-seen tasks. Note that each task's $\mathcal{D}_i$ is a distribution over $\mathcal{Z} := (\mathcal{X} \times \mathcal{Y})^{K+J}$. For some loss $\ell$, define the family of functions $\mathcal{F}(\mathcal{Z}) := \mathcal{F} := \{\hat{f}(w - \alpha \nabla \hat{f}(w, D^{train}), D^{test}) : w \in \mathcal{W}\}$, where $(D^{train}, D^{test}) \in \mathcal{Z}$ and $\hat{f}(w, D)$ is the average loss of $w$ on the samples in $D$. The Rademacher complexity of $\mathcal{F}$ on $m_i$ samples $\{(D_j^{train}, D_j^{test})\}_{j=1}^{m_i} =: \mathbf{D}$ drawn i.i.d. from $\mathcal{D}_i$ is then

$$\mathfrak{R}_{m_i}^i(\mathcal{F}) = \mathbb{E}_{\mathbf{D}\sim(\mathcal{D}_i)^{m_i}} \mathbb{E}_{\epsilon_j}\left[\sup_{w \in \mathcal{W}} \frac{1}{m_i}\sum_{j=1}^{m_i} \epsilon_j \hat{f}_{i,j}(w - \alpha \nabla \hat{f}_{i,j}(w, D_{i,j}^{train}), D_{i,j}^{test})\right], \qquad (11)$$

where the $\epsilon_j$'s are Rademacher random variables. Recall that the empirical loss of the model $w$ on the $i$-th task is $\hat{F}_i(w)$, defined in (2). By a standard Rademacher complexity bound, one can bound the analogous expected loss $F_i(w)$ with high probability over the choice of task instances.

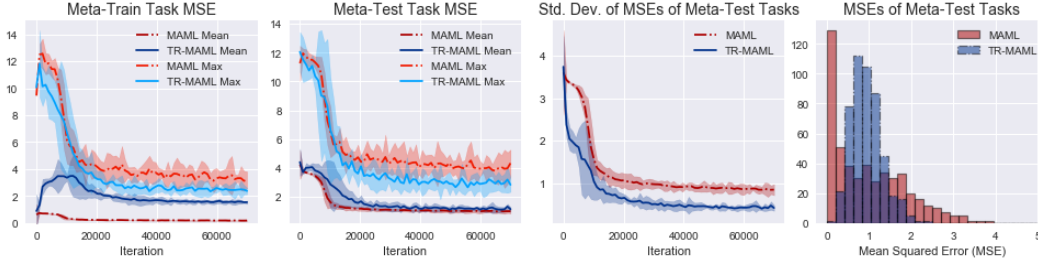

Figure 1: Meta-training and meta-test task MSE statistics vs the number of meta-training iterations for $K = 5$, with 95% confidence intervals shaded over 5 trials. The rightmost plot shows the number of meta-test tasks with average MSE within particular intervals for a sample trial. TR-MAML outperforms MAML on the worst-case regression task, and performs more uniformly across all tasks.

**Proposition 1.** *Suppose Assumption 1 holds, then with probability at least* $1 - \delta$ *for any* $\delta > 0$,

$$F_i(w) \coloneqq \mathbb{E}_{(D_{i,j}^{train}, D_{i,j}^{test}) \sim \mathcal{D}_i}[\hat{f}_{i,j}(w - \alpha \nabla \hat{f}_{i,j}(w, D_{i,j}^{train}), D_{i,j}^{test})] \leq \hat{F}_i(w) + 2\mathfrak{R}_{m_i}^i(\mathcal{F}) + \hat{B}\sqrt{\frac{\log 1/\delta}{2m_i}}$$

Next, let $w^*$ be the optimal solution to the TR-MAML meta-training objective (3). Suppose a new task is drawn with distribution $\mathcal{D}_{n+1}$, and suppose that $\mathcal{D}_{n+1} = \sum_{i=1}^n a_i \mathcal{D}_i$ for some $a \in \Delta_n$. Then the loss $F_{n+1}(w)$ is a convex combination of the losses on the meta-training tasks, yielding

**Theorem 4.** *For a new task with distribution* $\mathcal{D}_{n+1}$, *if* $\mathcal{D}_{n+1} = \sum_{i=1}^n a_i \mathcal{D}_i$ *for* $a \in \Delta_n$, *then with probability at least* $1 - \delta$ *for any* $\delta > 0$,

$$F_{n+1}(w^*) \leq \min_{w \in \mathcal{W}} \max_{p \in \Delta_n} \sum_{i=1}^n p_i \hat{F}_i(w) + 2a_i \mathfrak{R}_{m_i}^i(\mathcal{F}) + a_i \hat{B}\sqrt{\frac{\log(n/\delta)}{2m_i}} \tag{12}$$

Theorem 4 shows that the min-max meta-training solution leverages the diversity of the meta-training tasks to generalize across their full convex hull, not just a local neighborhood of the solution.

## 6 Experimental Results

**Experimental Setup:** Our experiments study whether minimizing the maximum task loss during meta-training leads to a more task-robust solution compared to MAML in few-shot sinusoid regression and image classification settings. Recall that our setting consists of a collection of tasks, with each task having a number of task-instances. In practice, the datasets we consider could have an exceedingly large number of tasks, thus rendering it computationally infeasible for us to conduct experiments. For instance, consider few-shot image classification on the Omniglot dataset, which is composed of images of characters from various alphabets. Suppose that we wish to do 5-way classification, meaning that there are images from 5 classes (characters) in each few-shot classification problem. In this setting, a task is a set of 5 particular classes (characters). There are 1200 meta-training characters in Omniglot, which would lead to $\binom{1200}{5}$ (around $2 \times 10^{13}$) total tasks for meta-training. Thus for for computational tractability, we reduce the number of tasks by clustering in a problem-dependent manner, resulting in the number of tasks ranging from few tens to hundred, as detailed below.

**Sinusoid Regression.** In the popular sinusoid regression experiment [12], each task instance is a sinusoid regression problem in which the target is a sine function on $[-5, 5] \subset \mathbb{R}$ with amplitude $a \in [0.1, 5]$ and phase $b \in [0, 2\pi]$. The learner has $K$ samples $\{(x_i, a\sin(x_i - b))\}_{i=1}^K$, where each $x_i$ is uniformly sampled from $[-5, 5]$, and tries to find a function that closely approximates $a\sin(x - b)$ in terms of mean squared error (MSE). Typically the meta-training and meta-testing distributions are identical, and are such that amplitudes are drawn uniformly from $[0.1, 5]$ and phases uniformly from $[0, 2\pi]$. Here we experiment with a distributional shift between meta-training and meta-testing in which a large number of easy task instances and a small number of hard task instances are accessible for meta-training, and the resulting initialization is evaluated on all tasks in the space. In particular, we assume that sine functions of all phases but with amplitudes only in the intervals $[0.1, 1.05]$ (easy tasks) and $[4.95, 5]$ (hard tasks) are available for meta-training. The sinusoids with larger amplitudes are harder targets because they are less smooth and have larger magnitudes, meaning poor approximations are generally punished more severely in terms of MSE. Empirically we find that phase has little effect on the hardness of a target.

Table 1: Sinusoid regression results showing MSE statistics across the 490 meta-test tasks, with 95% confidence intervals over 5 random trials.

| $K$ | Algorithm | Mean | Worst | Std. Dev. |
|---|---|---|---|---|
| 5 | MAML | $\mathbf{1.02 \pm 0.10}$ | $3.89 \pm 0.83$ | $0.88 \pm 0.14$ |
| | TR-MAML | $1.09 \pm 0.08$ | $\mathbf{2.82 \pm 0.35}$ | $\mathbf{0.43 \pm 0.03}$ |
| 10 | MAML | $\mathbf{0.66 \pm 0.16}$ | $2.57 \pm 0.70$ | $0.54 \pm 0.13$ |
| | TR-MAML | $0.77 \pm 0.11$ | $\mathbf{1.68 \pm 0.43}$ | $\mathbf{0.25 \pm 0.08}$ |

Table 2: Omniglot $N$-way, $K$-shot classification accuracies, with 95% confidence intervals over 3 random trials.

| $(N, K)$ | Algorithm | Meta-training Alphabets | | Meta-testing Alphabets | | |
|---|---|---|---|---|---|---|
| | | Weighted Mean | Worst | Weighted Mean | Worst | Std. Dev. |
| (5,1) | MAML | $\mathbf{98.4 \pm .2}$ | $82.4 \pm 1.1$ | $\mathbf{93.5 \pm .2}$ | $82.5 \pm .2$ | $3.84 \pm .1$ |
| | TR-MAML | $97.4 \pm .6$ | $\mathbf{95.0 \pm 0.3}$ | $93.1 \pm 1.1$ | $\mathbf{85.3 \pm 1.9}$ | $\mathbf{3.50 \pm .3}$ |
| (20,1) | MAML | $\mathbf{99.2 \pm .1}$ | $33.9 \pm 3.0$ | $67.6 \pm 2.0$ | $49.7 \pm 3.5$ | $9.10 \pm .1$ |
| | TR-MAML | $92.2 \pm .8$ | $\mathbf{82.4 \pm 2.1}$ | $\mathbf{74.3 \pm 1.4}$ | $\mathbf{58.4 \pm 1.8}$ | $\mathbf{8.70 \pm .5}$ |

We partition $[0.1, 5]$ into 490 disjoint subintervals of length 0.01, and define a task as the uniform distribution over all task instances with target amplitude in a particular subinterval. Thus, there are 95 easy and 5 hard meta-training tasks. We assume each task has the same number of instances available, so both MAML and TR-MAML sample phases uniformly from $[0, 2\pi]$ and amplitudes uniformly from $[0.1, 1.05] \cup [4.95, 5]$. The meta-test distribution is the uniform distribution across the full space of amplitudes and phases. Both algorithms use one SGD step as the inner learning algorithm, and the same fully-connected network architecture as in [12] for the learning model.

Figure 1 shows the convergence trajectories of MAML and TR-MAML when $K = 5$. Each plot entails estimating the current model's MSE on each task by sampling 5,000 task instances across all 100 meta-training tasks (for an average of 50 instances per task), and separately across all 490 meta-testing tasks. The leftmost plot shows the average and maximum MSE over each of the 100 meta-training tasks' estimated MSE vs the number of iterations, and the middle-left plot shows the same statistics over the 490 meta-testing tasks. During meta-training, TR-MAML sacrifices average for worst-case task performance. However, its focus on task-robustness yields more uniform performance across all tasks, allowing TR-MAML to outperform MAML on the hardest meta-test tasks while nearly matching MAML's average performance after the distribution shift. TR-MAML's more uniform performance for $K = 5$ is captured in the middle-right plot of Figure 1, which shows the standard deviation across the meta-testing task MSEs vs the number of iterations, and the rightmost plot of Figure 1, a histogram of the average MSEs among the 490 meta-test tasks. Table 1 tells a similar story for the $K \in \{5, 10\}$-shot cases by giving the average, maximum, and standard deviation of the MSEs among the 490 meta-test tasks after full meta-training, where the statistics are again empirical averages over 5,000 task instances.

**Image Classification.** In few-shot image classification, the task instances are $N$-way, $K$-shot classification problems, where $N$ is the number of classes and $K$ is the number of labeled samples from each class that are available to the learner. After updating the model based on these $NK$ samples, the model is evaluated on $J$ samples from each class. As discussed earlier, in standard few-shot image classification experiments, each individual $N$-way, $K$-shot classification problem is sometimes considered a 'task', leading to an intractably large number of tasks in our setting. Instead, we consider a more practical definition of a task as a set of $N$-way, $K$-shot classification problems (task instances) sharing similar properties (e.g. all $N$ characters belong to the same alphabet in the Omniglot experiment discussed below). Thus, a task instance is an individual $N$-way, $K$-shot classification problem, equivalent to the definition of a 'task' as used in other works.

We experiment in this setting using the Omniglot [19] and *mini*-ImageNet [37] datasets. For both datasets, we use the corresponding 4-layer CNN used in the original MAML paper [12]. Omniglot contains 1623 handwritten characters from 50 alphabets, with 20 examples per character. In order to establish an environment with a tractable number of tasks, we define each task as the uniform

Table 3: *mini*-ImageNet 5-way, 1-shot accuracies, with 95% confidence intervals over 3 random trials.

| $(N, K)$ | Algorithm | Eight Meta-Training Tasks | | Four Meta-Testing Tasks | |
|---|---|---|---|---|---|
| | | Weighted Mean | Worst | Weighted Mean | Worst |
| (5,1) | MAML | $\mathbf{70.1 \pm 2.2}$ | $48.0 \pm 4.5$ | $46.6 \pm .4$ | $44.7 \pm .7$ |
| | TR-MAML | $63.2 \pm 1.3$ | $\mathbf{60.7 \pm 1.6}$ | $\mathbf{48.5 \pm .6}$ | $\mathbf{45.9 \pm .8}$ |

distribution over all task instances composed of characters from *one particular alphabet*. Note that as a result, we sample the same fine-grained task instances, of characters all belonging to the same alphabet, as those recommended to use to evaluate meta-learning models on Omniglot in [36]. We use the same (meta-) train/validation/test splits as in [36]. There are $n = 25$ alphabets, i.e., tasks, for meta-training and 20 for meta-testing. Suppose there are $Z_i$ characters in the $i$-th alphabet, then the number of task instances that may be drawn from the $i$-th task is proportional to $\binom{Z_i}{N}$, since every character has the same number of samples. These proportions define the empirical distribution over the 25 meta-training tasks, so during meta-training MAML samples task instances by first selecting the $i$-th alphabet with probability proportional to $\binom{Z_i}{N}$, then uniformly samples an $N$-way, $K$-shot classification problem from the available data in alphabet $i$. Conversely, TR-MAML first samples an alphabet uniformly, then samples an $N$-way, $K$-shot problem uniformly from that alphabet.

After 60,000 meta-training iterations, we evaluate the models yielded by MAML and TR-MAML on 5,000 $N$-way, $K$-shot classification problems from the 20 meta-test alphabets, as well as 5,000 problems from the meta-training alphabets. Table 3 shows statistics taken over the average accuracy on task instances from each alphabet for different values of $N$ and $K$. First note that TR-MAML improves on MAML's worst-case task performance in all cases. Regarding mean performance, 'Weighted Mean' is the uniform average over task instances (i.e. is the surrogate for the expected loss over tasks given in Equation 1), and weighs the average accuracy on each task (alphabet) by the number of instances it contains. MAML aims to minimize this metric, and always outperforms TR-MAML on it. TR-MAML's improved 'Weighted Mean' performance at meta-test time in the $N = 20$ case shows that TR-MAML can generalize better than MAML because it prioritizes performance on all the meta-training tasks, whereas MAML may overfit to the most frequent ones. Observe that the empirical distribution of meta-training alphabets becomes more skewed as $N$ increases, causing MAML to focus on a smaller subset of the meta-training alphabets and further disregard worst-case alphabet performance, thus leading to worse generalization.

For *mini*-ImageNet, we split the 100 image classes into two subsets: 64 classes used for meta-training, and the remaining 36 for meta-testing, according to standard procedure [31] with the meta-validation classes used for meta-testing. We create tasks as follows: we randomly group the 64 meta-training classes into 8 meta-train tasks, with the numbers of classes/task being $\{6, 7, 7, 8, 8, 9, 9, 10\}$. Likewise, the 36 meta-test classes are randomly split into 4 tasks, each with 9 classes/task. Each task instance is constructed by sampling 1 image each from 5 distinct classes within a task: thus, this is 5-way 1-shot problem. We meta-train for 60,000 iterations with a batch size of 2 task instances, and 5 steps of gradient descent for local adaptation. Our results show the Weighted Mean accuracy (i.e. average case over task instances) and the worst-case performance (i.e. worst accuracy over the tasks). The first two columns are generated by testing on *new task instances* from the meta-training classes; the second two columns are generated by testing on task instances from the previously unseen meta-test classes. Again we see improved worst-case performance for TR-MAML compared to MAML, and improved mean performance is likely due to TR-MAML learning a more uniform model across the meta-training tasks.

**Concluding Remarks:** We propose TR-MAML[1], a MAML variant, that focuses on optimizing for robustness across tasks through a min-max formulation instead of an average case formulation. Our setting thus enables the model to provide reasonable performance even on hard and rarely seen tasks. However, shifting the focus to the worst-case does not come for free, as the model might suffer performance degradation in the average-case if some tasks are sufficiently outlying. Thus, the model that one would use in practice needs to be chosen appropriately depending on the deployment conditions and desired behavior.

## Broader Impact

Our work presents a formulation for learning how to learn optimally on the worst-case task from some environment. Although this formulation has no immediate societal consequences, it provides a novel framework for developing realizable meta-learning systems that are robust across all tasks. Such systems are necessary for many applications; one can think of few-shot fingerprint recognition in security systems, one-shot imitation learning for assembly line machines, and few-shot fraud detection as just a few examples. Moreover, systems that treat performance on all tasks equally despite disparities in the amount of data available for each task are critical for fairness in settings where tasks are correlated with people from a particular instance of a protected class such as race or gender. One weakness of our formulation is that it is not robust against adversarial tasks, but in settings where some tasks may be adversarial, our formulation may be modified to optimize the worst-case loss among the percentage of tasks known to be non-adversarial, the analysis of which we leave for future work.

## Acknowledgments and Disclosure of Funding

This work was partially supported by ONR Grant N00014-19-1-2566, NSF Grant SATC 1704778, NSF Grant CCF-2007668, and ARO grant W911NF-17-1-0359.

## Footnotes

[1]The code for TR-MAML is available at: `https://github.com/lgcollins/tr-maml`.

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
