[Supplementary Material]

<h1 style="text-align:center">Supplementary Material</h1>

## A    Formal Statement of Assumptions

**Assumption 1** For all $i \in [n]$ and $j \in [m_i]$, $\hat{f}_{i,j}(\cdot, D_{i,j}^{\text{test}})$ satisfy the following:

1. $\hat{B}$-boundedness: $\exists \hat{B} \in \mathbb{R}$ s.t. $\forall w \in \mathcal{W}$, $|\hat{f}_{i,j}(w, D_{i,j}^{\text{test}})| \leq \hat{B}$.

2. $\hat{L}$-Lipschitz continuity: $\exists \hat{L} \in \mathbb{R}$ s.t. $\forall u, v \in \mathcal{W}$, $|\hat{f}_{i,j}(u, D_{i,j}^{\text{test}}) - \hat{f}_{i,j}(v, D_{i,j}^{\text{test}})| \leq \hat{L}\|u - v\|_2$.

Furthermore, each $\hat{f}_{i,j}(\cdot, D_{i,j}^{\text{train}})$ satisfies the following:

1. Hessian eigenvalue lower bound: $\exists \hat{H} \in \mathbb{R}$ s.t. $\forall w \in \mathcal{W}$, $\lambda_{\min}(\nabla^2 \hat{f}_{i,j}(w, D_{i,j}^{\text{train}})) \geq -\hat{H}$.

**Assumption 2** For all $i \in [n]$ and $j \in [m_i]$, $\hat{f}_i(\cdot, D_{i,j}^{\text{train}})$ satisfies the following:

1. $\hat{M}$-smoothness: $\exists \hat{M} \in \mathbb{R}$ s.t. $\forall u, v \in \mathcal{W}$, $\|\nabla \hat{f}_{i,j}(u, D_{i,j}^{\text{train}}) - \nabla \hat{f}_{i,j}(v, D_{i,j}^{\text{train}})\|_2 \leq \hat{M}\|u - v\|_2$.

2. $\hat{\rho}$-Hessian-Lipschitz continuity: $\exists \hat{\rho} \in \mathbb{R}$ s.t. $\forall u, v \in \mathcal{W}$, $|\nabla^2 \hat{f}_{i,j}(u, D_{i,j}^{\text{train}}) - \nabla^2 \hat{f}_{i,j}(v, D_{i,j}^{\text{train}})| \leq \hat{\rho}\|u - v\|_2$.

## B    Proof of Lemma 1

### B.1    Unbiasedness

*Proof.* Recall that $\hat{g}_w(w, p)$ is computed as follows:

$$\hat{g}_w(w, p) = \frac{n}{C}\sum_{k=1}^{C} p_{i_k}(I - \alpha\nabla^2 \hat{f}_{i_k, j_k}(w, D_{i_k, j_k}^{\text{train}}))\nabla \hat{f}_{i_k, j_k}(w - \alpha\nabla \hat{f}_{i_k, j_k}(w, D_{i_k, j_k}^{\text{train}}), D_{i, j_i}^{\text{test}})$$

Thus we have

$\mathbb{E}[\hat{g}_w(w, p)]$

$= \mathbb{E}_{\{(i_k, j_k)\}_k}\left[\frac{n}{C}\sum_{k=1}^{C} p_{i_k}(I - \alpha\nabla^2 \hat{f}_{i_k, j_k}(w, D_{i_k, j_k}^{\text{train}}))\nabla \hat{f}_{i_k, j_k}(w - \alpha\nabla \hat{f}_{i_k, j_k}(w, D_{i_k, j_k}^{\text{train}}), D_{i_k, j_k}^{\text{test}})\right]$

$\stackrel{a}{=} \mathbb{E}_{\{i_k\}_k}\left[\frac{n}{C}\sum_{k=1}^{C}\mathbb{E}_{\{j_k\}_k}\left[p_{i_k}(I - \alpha\nabla^2 \hat{f}_{i_k, j_k}(w, D_{i_k, j_k}^{\text{train}}))\nabla \hat{f}_{i_k, j_k}(w - \alpha\nabla \hat{f}_{i_k, j_k}(w, D_{i_k, j_k}^{\text{train}}), D_{i_k, j_k}^{\text{test}}) \mid \{i_k\}_k\right]\right]$

$= \mathbb{E}_{\{i_k\}_k}\left[\frac{n}{C}\sum_{k=1}^{C}\frac{p_{i_k}}{m_{i_k}}\sum_{j=1}^{m_{i_k}}(I - \alpha\nabla^2 \hat{f}_{i_k, j}(w, D_{i_k, j}^{\text{train}}))\nabla \hat{f}_{i_k, j}(w - \alpha\nabla \hat{f}_{i_k, j}(w, D_{i_k, j}^{\text{train}}), D_{i_k, j}^{\text{test}}) \mid \mathcal{C}\right]$

$\stackrel{b}{=} \sum_{i=1}^{n}\frac{p_i}{m_i}\sum_{j=1}^{m_i}(I - \alpha\nabla^2 \hat{f}_{i,j}(w, D_{i,j}^{\text{train}}))\nabla \hat{f}_{i,j}(w - \alpha\nabla \hat{f}_{i,j}(w, D_{i,j}^{\text{train}}), D_{i,j}^{\text{test}})$

$= g_w(w, p) \hfill (13)$

where $a$ follows from the Law of iterated Expectation and $b$ follows because each index $i_k$ is selected with probability $1/n$. A similar computation shows that $\mathbb{E}[\hat{g}_p(w, p)] = g_p(w, p)$.  $\square$

### B.2    Bounded Second Moments

*Proof.* First we show the bound on $\mathbb{E}[\|\hat{g}_p(w, p)\|_2^2]$. Recall that $\hat{g}_p(w, p) = \sum_{k=1}^{C}\frac{n}{C}\hat{f}_{i_k}(w - \alpha\nabla \hat{f}_{i_k}(w, D_{i_k, j_k}^{\text{train}}), D_{i_k, j_k}^{\text{test}})e_{i_k}$. Let $c_i$ be the number of times index $i$ appears in $\{i_k\}_{k=1}^{C}$.

Then, noting that each $c_i$ is a binomial random variable with success probability $\frac{1}{n}$ and $C$ trials, we have

$$\mathbb{E}[\|\hat{g}_p(w,p)\|_2^2] = \sum_{i=1}^{n} \mathbb{E}[(\hat{g}_p(w,p)_i)^2]$$

$$= \sum_{i=1}^{n} \mathbb{E}[(\sum_{k\in[C]:i_k=i} \frac{n}{C}\hat{f}_i(w-\alpha\nabla\hat{f}_i(w,D_{i,j_k}^{\text{train}}),D_{i,j_k}^{\text{test}}))^2]$$

$$\leq \frac{n^2}{C^2}\sum_{i=1}^{n}\mathbb{E}[(\sum_{k\in[C]:i_k=i}\hat{B})^2] \tag{14}$$

$$= \frac{n^2}{C^2}\sum_{i=1}^{n}\mathbb{E}[(c_i\hat{B})^2]$$

$$= \frac{n^2}{C^2}\sum_{i=1}^{n}(\frac{C(n-1)}{n^2}+\frac{C^2}{n^2})\hat{B}^2$$

$$= \frac{n}{C}(n+C-1)\hat{B}^2 =: \hat{G}_p^2 \tag{15}$$

where (14) follows from Assumption 1. Next we bound $\mathbb{E}[\|\hat{g}_w(w,p)\|_2^2]$. Recall the definition of $\hat{g}_w(w,p)$:

$$\hat{g}_w(w,p) = \sum_{k=1}^{C}\frac{n}{C}p_{i_k}(I-\alpha\nabla^2\hat{f}_{i_k,j_k}(w,D_{i_k,j_k}^{\text{train}}))\nabla\hat{f}_{i_k,j_k}(w-\alpha\nabla\hat{f}_{i_k,j_k}(w,D_{i_k,j_k}^{\text{train}}),D_{i_k,j_k}^{\text{test}}) \tag{16}$$

We can write $\hat{g}_w(w,p) = \frac{n}{C}\sum_{k=1}^{C}X_k$ where $X_k$ is written as

$$X_k = p_{i_k}(I-\alpha\nabla^2\hat{f}_{i_k,j_k}(w,D_{i_k,j_k}^{\text{train}}))\nabla\hat{f}_{i_k,j_k}(w-\alpha\nabla\hat{f}_{i_k,j_k}(w,D_{i_k,j_k}^{\text{train}}),D_{i_k,j_k}^{\text{test}}) \tag{17}$$

We have

$$\mathbb{E}[\|X_k\|_2^2] = \frac{1}{n}\sum_{i=1}^{n}\frac{1}{m_i}\sum_{j=1}^{m_i}p_i^2\|(I-\alpha\nabla^2\hat{f}_{i,j}(w,D_{i,j}^{\text{train}}))\nabla\hat{f}_{i,j}(w-\alpha\nabla\hat{f}_{i,j}(w,D_{i,j}^{\text{train}}),D_{i,j}^{\text{test}})\|_2^2$$

$$\leq \frac{1}{n}\sum_{i=1}^{n}\frac{p_i^2}{m_i}\sum_{j=1}^{m_i}\|(I-\alpha\nabla^2\hat{f}_{i,j}(w,D_{i,j}^{\text{train}}))\|_2^2\|\nabla\hat{f}_{i,j}(w-\alpha\nabla\hat{f}_{i,j}(w,D_{i,j}^{\text{train}}),D_{i,j}^{\text{test}})\|_2^2$$

$$\tag{18}$$

$$\leq \frac{1}{n}\sum_{i=1}^{n}\frac{p_i^2}{m_i}\sum_{j=1}^{m_i}(1+\alpha\hat{H})^2\|\nabla\hat{f}_{i,j}(w-\alpha\nabla\hat{f}_i(w,D_{i,j}^{\text{train}}),D_{i,j}^{\text{test}})\|_2^2 \tag{19}$$

$$\leq \frac{1}{n}\sum_{i=1}^{n}\frac{p_i^2}{m_i}\sum_{j=1}^{m_i}(1+\alpha\hat{H})^2\hat{L}^2 \tag{20}$$

$$\leq \frac{1}{n}(1+\alpha\hat{H})^2\hat{L}^2 \tag{21}$$

where (18) follows by the Cauchy-Schwarz Inequality, (19) and (20) follow from Assumption 1, and (21) follows from the fact that $\sum_{i=1}^{n}p_i^2 \leq 1$. Thus we have

$$\mathbb{E}[\|\hat{g}_w(w,p)\|_2^2] = \mathbb{E}[\|\frac{n}{C}\sum_{k=1}^{C}X_k\|_2^2] \tag{22}$$

$$\leq \mathbb{E}[\frac{n^2}{C}\sum_{k=1}^{C}\|X_k\|_2^2] \tag{23}$$

$$\leq n(1+\alpha\hat{H})^2\hat{L}^2 \tag{24}$$

where (23) follows from the convexity of norms and Jensen's Inequality. $\square$

## C Proof of Theorem 1

*Proof.* We adapt the arguments from [24] to our nested gradients case. First observe that since each $\hat{F}_i(w)$ is convex, $\phi(w, p)$ is convex in $w$ and linear, thus concave, in $p$. Therefore we can write:

$$
\max_{p \in \Delta_n} \phi(w_T^{\mathbf{c}}, p) - \min_{w \in \mathcal{W}} \phi(w, p_T^{\mathbf{c}}) = \max_{p \in \Delta_n} \left\{ \phi(w_T^{\mathbf{c}}, p) - \min_{w \in \mathcal{W}} \phi(w, p_T^{\mathbf{c}}) \right\}
$$

$$
= \max_{p \in \Delta, w \in \mathcal{W}} \left\{ \phi(w_T^{\mathbf{c}}, p) - \phi(w, p_T^{\mathbf{c}}) \right\}
$$

$$
\leq \frac{1}{T} \max_{p \in \Delta, w \in \mathcal{W}} \left\{ \sum_{t=1}^{T} \phi(w^t, p) - \phi(w, p^t) \right\} \tag{25}
$$

where (25) follows from the convexity of $\phi$ in $w$ and the concavity of $\phi$ in $p$. Again using the convexity of $\phi$ in $w$ along with the linearity of $\phi$ in $p$, we have that for any $t \geq 1$,

$$
\phi(w^t, p) - \phi(w, p^t) = \phi(w^t, p) - \phi(w^t, p^t) + \phi(w^t, p^t) - \phi(w, p^t)
$$

$$
\leq \langle (p - p^t), \nabla_p \phi(w^t, p^t) \rangle + \langle (w^t - w), \nabla_w \phi(w^t, p^t) \rangle
$$

$$
= \langle (p - p^t), \hat{g}_p^t \rangle + \langle (w^t - w), \hat{g}_w^t \rangle
$$

$$
+ \langle (p - p^t), (\nabla_p \phi(w^t, p^t) - \hat{g}_p^t) \rangle + \langle (w^t - w), (\nabla_w \phi(w^t, p^t) - \hat{g}_w^t) \rangle
$$

Thus by rearranging terms and the subadditivity of $\max$,

$$
\max_{p \in \Delta, w \in \mathcal{W}} \left\{ \sum_{t=1}^{T} \phi(w^t, p) - \phi(w, p^t) \right\}
$$

$$
\leq \max_{p \in \Delta, w \in \mathcal{W}} \left\{ \sum_{t=1}^{T} \langle (p - p^t), \hat{g}_p^t \rangle + \langle (w^t - w), \hat{g}_w^t \rangle \right\}
$$

$$
+ \max_{p \in \Delta, w \in \mathcal{W}} \left\{ \sum_{t=1}^{T} \langle p, (\nabla_p \phi(w^t, p^t) - \hat{g}_p^t) \rangle + \langle w, (\hat{g}_w^t - \nabla_w \phi(w^t, p^t)) \rangle \right\}
$$

$$
- \left( \sum_{t=1}^{T} \langle p^t, (\nabla_p \phi(w^t, p^t) - \hat{g}_p^t) \rangle - \langle w^t, (\nabla_w \phi(w^t, p^t) - \hat{g}_w^t) \rangle \right) \tag{26}
$$

We bound the expectation of each of the above terms separately, starting with the first one. Note that since $2ab = a^2 + b^2 - (a - b)^2$, we have that for any $w \in \mathcal{W}$ and constant step size $\eta_w > 0$,

$$
\sum_{t=1}^{T} \langle (w^t - w), \hat{g}_w^t \rangle = \frac{1}{2} \sum_{t=1}^{T} \frac{1}{\eta_w} \|w^t - w\|_2^2 + \eta_w \|\hat{g}_w^t\|_2^2 - \frac{1}{\eta_w} \|w^t - \eta_w \hat{g}_w^t - w\|_2^2
$$

$$
\leq \frac{1}{2\eta_w} \sum_{t=1}^{T} \|w^t - w\|_2^2 + (\eta_w)^2 \|\hat{g}_w^t\|_2^2 - \|w^{t+1} - w\|_2^2 \tag{27}
$$

$$
= \frac{1}{2\eta_w} (\|w^1 - w\|_2^2 - \|w^{T+1} - w\|_2^2) + \frac{\eta_w}{2} \sum_{t=1}^{T} \|\hat{g}_w^t\|_2^2 \tag{28}
$$

$$
\leq \frac{1}{2\eta_w} \|w^1 - w\|_2^2 + \frac{\eta_w}{2} \sum_{t=1}^{T} \|\hat{g}_w^t\|_2^2
$$

$$
\leq \frac{2R_{\mathcal{W}}^2}{\eta_w} + \frac{\eta_w}{2} \sum_{t=1}^{T} \|\hat{g}_w^t\|_2^2 \tag{29}
$$

where (27) follows from the projection property and (28) is the result of the telescoping sum. Since (29) holds for all $w \in \mathcal{W}$ and its right hand side does does not depend on $w$, we can take the maximum over $w \in \mathcal{W}$ on the left hand side, and the expectation of both sides with respect to the stochastic gradients, to obtain

$$
\mathbb{E} \left[ \max_{w \in \mathcal{W}} \sum_{t=1}^{T} \langle (w^t - w), \hat{g}_w^t \rangle \right] \leq \frac{2R_{\mathcal{W}}^2}{\eta_w} + \frac{\eta_w T \hat{G}_w^2}{2} \tag{30}
$$

where $\hat{G}_w^2 = n(1 + \alpha\hat{H})^2\hat{L}^2$ is the bound on the second moment of the stochastic gradient with respect to $w$ given in Lemma 1. Using analogous arguments and noting that the radius of $\Delta_n$ is 1, we can show that

$$\mathbb{E}\left[\max_{p\in\Delta_n}\sum_{t=1}^T \langle (p - p^t), \hat{g}_p^t\rangle\right] \leq \frac{2}{\eta_p} + \frac{\eta_p T \hat{G}_p^2}{2} \tag{31}$$

where, again from Lemma 1, $\hat{G}_p^2 = \frac{n(n+C-1)\hat{B}^2}{C}$. Next, for the second term in (26), we can use the Cauchy-Schwarz Inequality and again the fact that $\max_{p\in\Delta_n}\|p\|_2 = 1$ to write

$$\max_{p\in\Delta_n}\sum_{t=1}^T \langle p, \nabla_p\phi(w^t, p^t) - \hat{g}_p^t\rangle = \max_{p\in\Delta_n}\langle p, \sum_{t=1}^T \nabla_p\phi(w^t, p^t) - \hat{g}_p^t\rangle$$

$$\leq \|\sum_{t=1}^T \nabla_p\phi(w^t, p^t) - \hat{g}_p^t\|_2 \tag{32}$$

Note from Lemma 1 that

$$\mathbb{E}[\|\nabla_p\phi(w^t, p^t) - \hat{g}_p^t\|_2^2] = \mathbb{E}[\|\hat{g}_p^t\|_2^2] - \|\nabla_p\phi(w^t, p^t)\|_2^2$$

$$\leq \hat{G}_p^2$$

for all $t \geq 1$. Define $\tilde{\sigma}_p^2$ such that $\mathbb{E}[\|\nabla_p\phi(w^t, p^t) - \hat{g}_p^t\|_2^2] \leq \tilde{\sigma}_p^2 \leq \hat{G}_p^2$ for all $t \geq 1$. Also note that because the batch selections are independent, the $\nabla_p\phi(w^t, p^t) - \hat{g}_p^t$ terms are uncorrelated random variables with mean 0. Using this fact combined with the definition of $\tilde{\sigma}_p^2$, we obtain

$$\mathbb{E}[\|\sum_{t=1}^T \nabla_p\phi(w^t, p^t) - \hat{g}_p^t\|_2]^2 \leq \mathbb{E}[\|\sum_{t=1}^T \nabla_p\phi(w^t, p^t) - \hat{g}_p^t\|_2^2]$$

$$= \mathbb{E}[\sum_{t=1}^T \|\nabla_p\phi(w^t, p^t) - \hat{g}_p^t\|_2^2]$$

$$\leq T\tilde{\sigma}_p^2$$

which implies that $\mathbb{E}[\|\sum_{t=1}^T \nabla_p\phi(w^t, p^t) - \hat{g}_p^t\|_2] \leq \sqrt{T}\tilde{\sigma}_p$. Using this relation after taking the expectation of both sides of (32) yields

$$\mathbb{E}\left[\max_{p\in\Delta_n}\sum_{t=1}^T \langle p, \nabla_p\phi(w^t, p^t) - \hat{g}_p^t\rangle\right] \leq \sqrt{T}\tilde{\sigma}_p \tag{33}$$

Using similar arguments and the analogous definition of $\tilde{\sigma}_w^2$, with this time using $R_\mathcal{W}$ to bound $\max_{w\in\mathcal{W}}\|w\|_2$ after the analogous Cauchy-Schwarz step as in 32, we have

$$\mathbb{E}\left[\max_{w\in\mathcal{W}}\sum_{t=1}^T \langle w, \hat{g}_w^t - \nabla_w\phi(w^t, p^t)\rangle\right] \leq R_\mathcal{W}\sqrt{T}\tilde{\sigma}_w \tag{34}$$

For the third and final term in (26), note that by the Law of Iterated Expectations and the unbiasedness of the stochastic gradients, we have that for any $t \geq 1$,

$$\mathbb{E}[\langle p^t, (\nabla_p\phi(w^t, p^t) - \hat{g}_p^t)\rangle - \langle w^t, (\nabla_w\phi(w^t, p^t) - \hat{g}_w^t)\rangle]$$

$$= \mathbb{E}\left[\mathbb{E}\left[\langle p^t, (\nabla_p\phi(w^t, p^t) - \hat{g}_p^t)\rangle - \langle w^t, (\nabla_w\phi(w^t, p^t) - \hat{g}_w^t)\rangle | w^t, p^t\right]\right]$$

$$= 0$$

Recalling (25) and (26), by combining the bounds on each of the terms and dividing by $T$, we obtain

$$\mathbb{E}\left[\max_{p\in\Delta_n}\phi(w_T^C, p) - \min_{w\in\mathcal{W}}\phi(w, p_T^C)\right] \leq \frac{2R_\mathcal{W}^2}{\eta_w T} + \frac{\eta_w\hat{G}_w^2}{2} + \frac{2}{\eta_p T} + \frac{\eta_p\hat{G}_p^2}{2} + \frac{R_\mathcal{W}\tilde{\sigma}_w}{\sqrt{T}} + \frac{\tilde{\sigma}_p}{\sqrt{T}} \tag{35}$$

We minimize the above bound by setting the step sizes as

$$\eta_w = \frac{2R_\mathcal{W}}{\hat{G}_w\sqrt{T}}, \quad \eta_p = \frac{2}{\hat{G}_p\sqrt{T}} \tag{36}$$

to complete the proof, noting that $\tilde{\sigma}_w \leq \hat{G}_w$ and $\tilde{\sigma}_p \leq \hat{G}_p$. $\qquad\square$

# D Proof of Lemma 2

The result is a sample-approximation version of Theorem 1 in [13]. We include our version of the proof here for completeness.

*Proof.* Note that $\hat{F}_i(w)$ is the empirical average of the functions $\hat{f}_{i,j}(w - \alpha \nabla \hat{f}_{i,j}(w, D_{i,j}^{\text{train}}), D_{i,j}^{\text{test}})$ for $j = 1, \ldots, m_i$, so we can write $\hat{F}_i(w)$ as the empirical expectation over $j$:

$$\hat{\mathbb{E}}_j[\hat{f}_{i,j}(w - \alpha \nabla \hat{f}_{i,j}(w, D_{i,j}^{\text{train}}), D_{i,j}^{\text{test}})] := \frac{1}{m_i} \sum_{j=1}^{m_i} \hat{f}_{i,j}(w - \alpha \nabla \hat{f}_{i,j}(w, D_{i,j}^{\text{train}}), D_{i,j}^{\text{test}}) = \hat{F}_i(w) \quad (37)$$

Using this notation, we show the strong convexity of $\hat{F}_i$ when $\alpha < 1/M$ and each $\hat{f}_{i,j}(\cdot, D_{i,j}^{\text{test}})$ is $\mu$-strongly convex in addition to satisfying Assumption 1. We have

$$\begin{aligned}
&\|\nabla \hat{F}_i(u) - \nabla \hat{F}_i(v)\| \\
&= \|\hat{\mathbb{E}}_j\big[(I - \alpha \nabla^2 \hat{f}_{i,j}(u, D_{i,j}^{\text{train}}))\nabla \hat{f}_{i,j}(u - \alpha \nabla \hat{f}_{i,j}(u, D_{i,j}^{\text{train}}), D_{i,j}^{\text{test}}) \\
&\qquad - (I - \alpha \nabla^2 \hat{f}_{i,j}(v, D_{i,j}^{\text{train}}))\nabla \hat{f}_{i,j}(v - \alpha \nabla \hat{f}_{i,j}(v, D_{i,j}^{\text{train}}), D_{i,j}^{\text{test}})\big]\| \\
&= \|\hat{\mathbb{E}}_j\big[(I - \alpha \nabla^2 \hat{f}_{i,j}(u, D_{i,j}^{\text{train}}))\big(\nabla \hat{f}_{i,j}(u - \alpha \nabla \hat{f}_{i,j}(u, D_{i,j}^{\text{train}}), D_{i,j}^{\text{test}}) \\
&\qquad - \nabla \hat{f}_{i,j}(v - \alpha \nabla \hat{f}_{i,j}(v, D_{i,j}^{\text{train}}), D_{i,j}^{\text{test}})\big) - \Big((I - \alpha \nabla^2 \hat{f}_{i,j}(v, D_{i,j}^{\text{train}})) \\
&\qquad - (I - \alpha \nabla^2 \hat{f}_{i,j}(u, D_{i,j}^{\text{train}}))\big)\nabla \hat{f}_{i,j}(v - \alpha \nabla \hat{f}_{i,j}(v, D_{i,j}^{\text{train}}), D_{i,j}^{\text{test}})\big]\| \qquad (38) \\
&\geq \|\hat{\mathbb{E}}_j\big[(I - \alpha \nabla^2 \hat{f}_{i,j}(u, D_{i,j}^{\text{train}}))(\nabla \hat{f}_{i,j}(u - \alpha \nabla \hat{f}_{i,j}(u, D_{i,j}^{\text{train}}), D_{i,j}^{\text{test}}) \\
&\qquad - \nabla \hat{f}_{i,j}(v - \alpha \nabla \hat{f}_{i,j}(v, D_{i,j}^{\text{train}}), D_{i,j}^{\text{test}}))\big]\| \\
&\quad - \|\hat{\mathbb{E}}_j\big[((I - \alpha \nabla^2 \hat{f}_{i,j}(v, D_{i,j}^{\text{train}})) \\
&\qquad - (I - \alpha \nabla^2 \hat{f}_{i,j}(u, D_{i,j}^{\text{train}})))\nabla \hat{f}_{i,j}(v - \alpha \nabla \hat{f}_{i,j}(v, D_{i,j}^{\text{train}}), D_{i,j}^{\text{test}})\big]\| \\
&= \|\hat{\mathbb{E}}_j\big[(I - \alpha \nabla^2 \hat{f}_{i,j}(u, D_{i,j}^{\text{train}}))(\nabla \hat{f}_{i,j}(u - \alpha \nabla \hat{f}_{i,j}(u, D_{i,j}^{\text{train}}), D_{i,j}^{\text{test}}) \\
&\quad - \nabla \hat{f}_{i,j}(v - \alpha \nabla \hat{f}_{i,j}(v, D_{i,j}^{\text{train}}), D_{i,j}^{\text{test}}))\big]\| \\
&\quad - \alpha \|\hat{\mathbb{E}}_j\big[(\nabla^2 \hat{f}_{i,j}(u, D_{i,j}^{\text{train}}) - \nabla^2 \hat{f}_{i,j}(v, D_{i,j}^{\text{train}}))\nabla \hat{f}_{i,j}(v - \alpha \nabla \hat{f}_{i,j}(v, D_{i,j}^{\text{train}}), D_{i,j}^{\text{test}})\big]\| \qquad (39)
\end{aligned}$$

To lower bound the first term, we use the $\hat{M}$-smoothness of $\hat{f}_{i,j}(\cdot, D_{i,j}^{\text{train}})$, which implies that the minimum eigenvalue of $I - \alpha \nabla^2 \hat{f}_{i,j}(u, D_{i,j}^{\text{train}})$ is at least $1 - \alpha \hat{M}$ for all $u \in \mathcal{W}$. Thus,

$$\begin{aligned}
&\|\hat{\mathbb{E}}_j\big[(I - \alpha \nabla^2 \hat{f}_{i,j}(u, D_{i,j}^{\text{train}}))(\nabla \hat{f}_{i,j}(u - \alpha \nabla \hat{f}_{i,j}(u, D_{i,j}^{\text{train}}), D_{i,j}^{\text{test}}) \\
&\qquad - \nabla \hat{f}_{i,j}(v - \alpha \nabla \hat{f}_{i,j}(v, D_{i,j}^{\text{train}}), D_{i,j}^{\text{test}}))\big]\| \\
&\geq (1 - \alpha \hat{M})\|\hat{\mathbb{E}}_j\big[\nabla \hat{f}_{i,j}(u - \alpha \nabla \hat{f}_{i,j}(u, D_{i,j}^{\text{train}}), D_{i,j}^{\text{test}}) - \nabla \hat{f}_{i,j}(v - \alpha \nabla \hat{f}_{i,j}(v, D_{i,j}^{\text{train}}), D_{i,j}^{\text{test}})\big]\|
\end{aligned}$$
$$(40)$$

By the $\hat{\mu}$-strong convexity of $\hat{f}_{i,j}(\cdot, D_{i,j}^{\text{test}})$ and the triangle inequality, we have

$$\begin{aligned}
&\|\hat{\mathbb{E}}_j\big[\nabla \hat{f}_{i,j}(u - \alpha \nabla \hat{f}_{i,j}(u, D_{i,j}^{\text{train}}), D_{i,j}^{\text{test}}) - \nabla \hat{f}_{i,j}(v - \alpha \nabla \hat{f}_{i,j}(v, D_{i,j}^{\text{train}}), D_{i,j}^{\text{test}})\big]\| \\
&\qquad \geq \hat{\mu}\|\hat{\mathbb{E}}_j\big[u - \alpha \nabla \hat{f}_{i,j}(u, D_{i,j}^{\text{train}}) - (v - \alpha \nabla \hat{f}_{i,j}(v, D_{i,j}^{\text{train}}))\big]\| \\
&\qquad \geq \hat{\mu}\Big(\|u - v\| - \alpha \|\hat{\mathbb{E}}_j\big[\nabla \hat{f}_{i,j}(v, D_{i,j}^{\text{train}}) - \nabla \hat{f}_{i,j}(u, D_{i,j}^{\text{train}})\big]\|\Big) \\
&\qquad \geq \hat{\mu}\Big(\|u - v\| - \alpha \hat{\mathbb{E}}_j \|\nabla \hat{f}_{i,j}(v, D_{i,j}^{\text{train}}) - \nabla \hat{f}_{i,j}(u, D_{i,j}^{\text{train}})\|\Big) \\
&\qquad \geq \mu\Big(\|u - v\| - \alpha \hat{M}\|u - v\|\Big) \qquad (41)
\end{aligned}$$

where the second-to-last inequality follows from Jensen's Inequality and the last inequality follows from the $\hat{M}$-smoothness of each $\hat{f}_{i,j}(\cdot, D_{i,j}^{\text{train}})$ Next we upper bound the second term in (39). We have

$$\|\hat{\mathbb{E}}_j\big[(\nabla^2 \hat{f}_{i,j}(u, D_{i,j}^{\text{train}}) - \nabla^2 \hat{f}_{i,j}(v, D_{i,j}^{\text{train}}))\nabla \hat{f}_{i,j}(v - \alpha \nabla \hat{f}_{i,j}(v, D_{i,j}^{\text{train}}), D_{i,j}^{\text{test}})\big]\|$$

$$\leq \hat{\mathbb{E}}_j\big[\|(\nabla^2 \hat{f}_{i,j}(u, D_{i,j}^{\text{train}}) - \nabla^2 \hat{f}_{i,j}(v, D_{i,j}^{\text{train}}))\nabla \hat{f}_{i,j}(v - \alpha \nabla \hat{f}_{i,j}(v, D_{i,j}^{\text{train}}), D_{i,j}^{\text{test}})\|\big] \quad (42)$$

$$\leq \hat{\mathbb{E}}_j\big[\|(\nabla^2 \hat{f}_{i,j}(u, D_{i,j}^{\text{train}}) - \nabla^2 \hat{f}_{i,j}(v, D_{i,j}^{\text{train}}))\|\|\nabla \hat{f}_{i,j}(v - \alpha \nabla \hat{f}_{i,j}(v, D_{i,j}^{\text{train}}), D_{i,j}^{\text{test}})\|\big] \quad (43)$$

$$\leq \sqrt{\hat{\mathbb{E}}_j[\|\nabla^2 \hat{f}_{i,j}(u, D_{i,j}^{\text{train}}) - \nabla^2 \hat{f}_{i,j}(v, D_{i,j}^{\text{train}})\|^2]\hat{\mathbb{E}}_j[\|\nabla \hat{f}_{i,j}(v - \alpha \nabla \hat{f}_{i,j}(v, D_{i,j}^{\text{train}}), D_{i,j}^{\text{test}})\|^2]} \quad (44)$$

$$\leq \hat{L}\sqrt{\hat{\mathbb{E}}_j[\|\nabla^2 \hat{f}_{i,j}(u, D_{i,j}^{\text{train}}) - \nabla^2 \hat{f}_{i,j}(v, D_{i,j}^{\text{train}})\|^2]} \quad (45)$$

$$\leq \hat{L}\hat{\rho}\|u - v\| \quad (46)$$

where (42) follows from Jensen's Inequality, (43) and (44) follow from the Cauchy-Schwarz Inequality, (45) follows from the $\hat{L}$-Lipschitzness of $\hat{f}_{i,j}(v, D_{i,j}^{\text{test}})$ for all $j \in [m_i]$, and (46) follows from Assumption 1. Combining (39), (40), and (41) and (46) yields that $\hat{F}_i$ is $\tilde{\mu} := (\hat{\mu}(1 - \alpha \hat{M})^2 - \alpha \hat{L}\hat{\rho})$-strongly convex under the given conditions. $\qquad\square$

## E  Proof of Lemma 3

*Proof.* We show the smoothness of each $\hat{F}_i$ by upper bounding the norm of the difference of its gradients. Using (38) and the triangle inequality,

$$\|\nabla \hat{F}_i(u) - \nabla \hat{F}_i(v)\|$$
$$\leq \|\hat{\mathbb{E}}_j\big[(I - \alpha \nabla^2 \hat{f}_{i,j}(u, D_{i,j}^{\text{train}}))(\nabla \hat{f}_{i,j}(u - \alpha \nabla \hat{f}_{i,j}(u, D_{i,j}^{\text{train}}), D_{i,j}^{\text{test}}) - \nabla \hat{f}_{i,j}(v - \alpha \nabla \hat{f}_{i,j}(v, D_{i,j}^{\text{train}}), D_{i,j}^{\text{test}}))\big]\|$$
$$+ \|\hat{\mathbb{E}}_j\big[((I - \alpha \nabla^2 \hat{f}_{i,j}(v, D_{i,j}^{\text{train}})) - (I - \alpha \nabla^2 \hat{f}_{i,j}(u, D_{i,j}^{\text{train}})))\nabla \hat{f}_{i,j}(v - \alpha \nabla \hat{f}_{i,j}(v, D_{i,j}^{\text{train}}), D_{i,j}^{\text{test}})\big]\| \quad (47)$$

We consider the two terms in the right hand side of (47) separately. Denoting the first term as $\Xi$, we use Jensen's Inequality then the Cauchy-Schwarz Inequality twice, as in (43) and (44), to obtain

$$\Xi \leq \big(\hat{\mathbb{E}}_j\big[\|I - \alpha \nabla^2 \hat{f}_{i,j}(u, D_{i,j}^{\text{train}})\|^2\big]\hat{\mathbb{E}}_j\big[\|\nabla \hat{f}_{i,j}(u - \alpha \nabla \hat{f}_{i,j}(u, D_{i,j}^{\text{train}}), D_{i,j}^{\text{test}})$$
$$- \nabla \hat{f}_{i,j}(v - \alpha \nabla \hat{f}_{i,j}(v, D_{i,j}^{\text{train}}), D_{i,j}^{\text{test}})\|^2\big]\big)^{1/2}$$
$$\leq (1 + \alpha \hat{M})\sqrt{\hat{\mathbb{E}}_j\big[\|\nabla \hat{f}_{i,j}(u - \alpha \nabla \hat{f}_{i,j}(u, D_{i,j}^{\text{train}}), D_{i,j}^{\text{test}}) - \nabla \hat{f}_{i,j}(v - \alpha \nabla \hat{f}_{i,j}(v, D_{i,j}^{\text{train}}), D_{i,j}^{\text{test}})\|^2\big]} \quad (48)$$

where to obtain (48) we have used the $M$-smoothness of $\hat{f}_{i,j}$. Considering the term remaining inside the square root, we have

$$\hat{\mathbb{E}}_j\big[\|\nabla \hat{f}_{i,j}(u - \alpha \nabla \hat{f}_{i,j}(u, D_{i,j}^{\text{train}}), D_{i,j}^{\text{test}}) - \nabla \hat{f}_{i,j}(v - \alpha \nabla \hat{f}_{i,j}(v, D_{i,j}^{\text{train}}), D_{i,j}^{\text{test}})\|^2\big]$$

$$\leq \hat{M}^2 \hat{\mathbb{E}}_j\big[\|u - \alpha \nabla \hat{f}_{i,j}(u, D_{i,j}^{\text{train}}) - (v - \alpha \nabla \hat{f}_{i,j}(v, D_{i,j}^{\text{train}}))\|^2\big] \quad (49)$$

$$= \hat{M}^2 \hat{\mathbb{E}}_j\big[\|u - v\|^2 + 2\alpha(u - v)^T(\nabla \hat{f}_{i,j}(v, D_{i,j}^{\text{train}}) - \nabla \hat{f}_{i,j}(u, D_{i,j}^{\text{train}}))$$
$$+ \alpha^2 \|\nabla \hat{f}_{i,j}(u, D_{i,j}^{\text{train}}) - \nabla \hat{f}_{i,j}(v, D_{i,j}^{\text{train}})\|^2\big]$$

$$= \hat{M}^2 \bigg(\|u - v\|^2 + 2\alpha^2(u - v)^T \mathbb{E}_j[\nabla \hat{f}_{i,j}(u, D_{i,j}^{\text{train}}) - \nabla \hat{f}_{i,j}(v, D_{i,j}^{\text{train}})]$$

$$+ \alpha^2 \mathbb{E}_j[\|\nabla \hat{f}_{i,j}(u, D_{i,j}^{\text{train}}) - \nabla \hat{f}_{i,j}(v, D_{i,j}^{\text{train}})\|^2]\bigg)$$

$$\leq \hat{M}^2 \bigg(\|u - v\|^2 + 2\alpha \hat{M}\|u - v\|^2 + \alpha^2 \hat{M}^2\|u - v\|^2\bigg) \quad (50)$$

$$= \hat{M}^2 \bigg(1 + \alpha \hat{M}\bigg)^2 \|u - v\|^2 \quad (51)$$

where (50) follows from the $\hat{M}$-smoothness of $\hat{f}_{i,j}(\cdot, D_{i,j}^{\text{train}})$ and the Cauchy Schwarz Inequality. Thus we have

$$\Xi \le \hat{M}(1 + \alpha\hat{M})^2 \|u - v\| \tag{52}$$

Note that we have already upper bounded the second term in (47) in the previous lemma (see Equation (46)). Thus we have that the smoothness parameter of $\hat{F}_i$ is

$$\tilde{M}_i := \hat{M}(1 + \alpha\hat{M})^2 + \alpha\hat{L}\hat{\rho} \tag{53}$$

$\square$

# F   Proof of Lemma 4

*Proof.* We again use the shorthand $X_k$ as defined in Appendix (17), and we also define

$$X_{i,j} = p_i(I - \alpha\nabla^2 \hat{f}_{i,j}(w, D_{i,j}^{\text{train}}))\nabla \hat{f}_{i,j}(w - \alpha\nabla \hat{f}_{i,j}(w, D_{i,j}^{\text{train}}), D_{i,j}^{\text{test}}) \tag{54}$$

for a fixed $i \in [n]$ and $j \in [m_i]$. Note that $X_k$ is a random variable while $X_{i,j}$ is deterministic. Also observe that $\hat{g}_w(w, p) = \frac{1}{C}\sum_{k=1}^{C} nX_k$, that each $nX_k$ is an unbiased estimate of $g_w(w, p)$, and the $X_k$'s are independent. Using these facts, we have

$$\mathbb{E}[\|\hat{g}_w(w, p) - g_w(w, p)\|_2^2]$$

$$= \mathbb{E}[\|\frac{1}{C}\sum_{k=1}^{C} nX_k - g_w(w, p)\|_2^2] \tag{55}$$

$$= \frac{1}{C^2}\mathbb{E}[\sum_{k=1}^{C} \|nX_k - g_w(w, p)\|_2^2] \tag{56}$$

$$= \frac{1}{C}\mathbb{E}[\|nX_1 - g_w(w, p)\|_2^2] \tag{57}$$

$$= \frac{1}{nC}\sum_{i=1}^{n}\frac{1}{m_i}\sum_{j=1}^{m_i} \|nX_{i,j} - g_w(w, p)\|_2^2 \tag{58}$$

$$= \frac{1}{nC}\sum_{i=1}^{n}\frac{1}{m_i}\sum_{j=1}^{m_i} \|nX_{i,j} - np_i\nabla\hat{F}_i(w)\|_2^2 + \|np_i\nabla\hat{F}_i(w) - g_w(w, p)\|_2^2$$

$$- 2(nX_{i,j} - np_i\nabla\hat{F}_i(w))(np_i\nabla\hat{F}_i(w) - g_w(w, p)) \tag{59}$$

Consider

$$\sum_{j=1}^{m_i}(nX_{i,j} - np_i\nabla\hat{F}_i(w))(np_i\nabla\hat{F}_i(w) - g_w(w, p))$$

$$= (np_i\nabla\hat{F}_i(w) - g_w(w, p))\sum_{j=1}^{m_i}(nX_{i,j} - np_i\nabla\hat{F}_i(w))$$

$$= n(np_i\nabla\hat{F}_i(w) - g_w(w, p))\left[\sum_{j=1}^{m_i}\left(p_i(I - \alpha\nabla^2\hat{f}_i(w, D_{i,j}^{\text{train}}))\nabla\hat{f}_i(w - \alpha\nabla\hat{f}_i(w, D_{i,j}^{\text{train}}), D_{i,j}^{\text{test}})\right) - m_ip_i\nabla\hat{F}_i(w)\right]$$

$$= n(np_i\nabla\hat{F}_i(w) - g_w(w, p))\left[m_ip_i\nabla\hat{F}_i(w) - m_ip_i\nabla\hat{F}_i(w)\right]$$

$$= 0$$

Therefore we have

$$\mathbb{E}[\|\hat{g}_w(w,p) - g_w(w,p)\|_2^2]$$

$$= \frac{1}{nC}\sum_{i=1}^{n}\frac{1}{m_i}\sum_{j=1}^{m_i}\|nX_{i,j} - np_i\nabla\hat{F}_i(w)\|_2^2 + \|np_i\nabla\hat{F}_i(w) - g_w(w,p)\|_2^2$$

$$= \frac{n}{C}\sum_{i=1}^{n}\frac{1}{m_i}\sum_{j=1}^{m_i}\|p_iX_{i,j} - p_i\sum_{j'=1}^{m_i}X_{i,j'}\|_2^2 + \frac{n}{C}\sum_{i=1}^{n}\|p_i\nabla\hat{F}_i(w) - \frac{1}{n}\sum_{i'=1}^{n}p_{i'}\nabla\hat{F}_{i'}(w)\|_2^2$$

$$= \frac{n}{C}\sum_{i=1}^{n}\sigma_i^2 + \frac{n}{C}\sigma^2$$

$\square$

# G Proof of Theorem 2

**Proposition 2.** *Suppose Assumption 1 holds and $\mathcal{W} = \mathbb{R}^d$. Let $\eta_w^t$ and $\eta_p^t$ be constant over all $t$, denoted by $\eta_w$ and $\eta_p$, respectively, where $\eta_w < (2/\tilde{M})$. Let $(w_T^\tau, p_T^\tau)$ be the solution returned by Algorithm 1 after $T$ iterations. Then,*

$$\mathbb{E}[\|\nabla_w\phi(w_T^\tau, p_T^\tau)\|_2^2] \leq \frac{2(\phi(w^1, p^1) + B)}{T(2\eta_w - \eta_w^2\tilde{M})} + \frac{4\eta_p\sqrt{n}B\hat{G}_p}{(2\eta_w - \eta_w^2\tilde{M})} + \frac{\eta_w\tilde{M}\sigma_w^2}{(2 - \eta_w\tilde{M})},$$

$$\mathbb{E}\left[\phi(w_T^\tau, p_T^\tau)\right] \geq \max_{p\in\Delta_n}\left\{\mathbb{E}\left[\phi(w_T^\tau, p)\right]\right\} - \frac{1}{\eta_p T} - \frac{\eta_p\hat{G}_p^2}{2}$$

*where $\hat{G}_p^2 = n(n + C - 1)\hat{B}^2/C$.*

*Proof.* Note that

$$\mathbb{E}[\|\nabla_w\phi(w_T^\tau, p_T^\tau)\|_2^2] = \mathbb{E}\left[\mathbb{E}_\tau[\|\nabla_w\phi(w_T^\tau, p_T^\tau)\|_2^2]\right] \tag{60}$$

$$= \mathbb{E}\left[\frac{1}{T}\sum_{t=1}^{T}\|\nabla_w\phi(w^t, p^t)\|_2^2\right] \tag{61}$$

$$= \frac{1}{T}\sum_{t=1}^{T}\mathbb{E}\left[\|g_w^t\|^2\right] \tag{62}$$

where the un-subscripted expectation in the right hand sides of (60) and (61) is over the stochastic gradients which determine the sequence $\{(w^t, p^t)\}_t$. Thus to show the bound on $\mathbb{E}[\|\nabla_w\phi(w_T^\tau, p_T^\tau)\|_2^2]$ in Proposition 2, we bound the right hand side of (62). To do so we borrow ideas from the proof of Theorem 1 in [28]. First recall that by Lemma 3, $\hat{F}_i$ is $\tilde{M}$-smooth for each $i \in \{1, ..., n\}$. Then for any $u, v \in \mathcal{W}$,

$$\hat{F}_i(u) \leq \hat{F}_i(v) + \nabla\hat{F}_i(v)^T(u - v) + \frac{\tilde{M}}{2}\|u - v\|^2 \tag{63}$$

Conditioned on the history up to iteration $t$, denoted by $\mathcal{F}^t$, the above equation implies

$$\mathbb{E}\left[\sum_{i=1}^{n}p_i^t\hat{F}_i(w^{t+1})|\mathcal{F}^t\right]$$

$$\leq \mathbb{E}\left[\sum_{i=1}^{n}p_i^t\hat{F}_i(w^t) + \left(\nabla_w\sum_{i=1}^{n}p_i^t\hat{F}_i(w^t)\right)^T(w^{t+1} - w^t) + \frac{\tilde{M}}{2}\|w^{t+1} - w^t\|^2|\mathcal{F}^t\right] \tag{64}$$

Note that $\nabla_w \sum_{i=1}^n p_i^t \hat{F}_i(w^t) = g_w^t$ and $w^{t+1} - w^t = -\eta_w \hat{g}_w^t$. Thus, we have

$$\mathbb{E}\left[\sum_{i=1}^n p_i^t \hat{F}_i(w^{t+1})|\mathcal{F}^t\right] \leq \mathbb{E}\left[\sum_{i=1}^n p_i^t \hat{F}_i(w^t) - \eta_w (g_w^t)^T \hat{g}_w^t + \frac{\tilde{M}}{2}\eta_w^2 \|\hat{g}_w^t\|^2|\mathcal{F}^t\right] \tag{65}$$

$$= \sum_{i=1}^n p_i^t \hat{F}_i(w^t) - \eta_w \|g_w^t\|^2 + \frac{\tilde{M}}{2}\eta_w^2 \left(\|g_w^t\|^2 + \mathbb{E}\left[\|\hat{g}_w^t - g_w^t\|^2|\mathcal{F}^t\right]\right) \tag{66}$$

where (66) follows because $\hat{g}_w^t$ is an unbiased estimate of $g_w^t$. Using Lemma 4, we have

$$\mathbb{E}\left[\sum_{i=1}^n p_i^t \hat{F}_i(w^{t+1})|\mathcal{F}^t\right] \leq \sum_{i=1}^n p_i^t \hat{F}_i(w^t) - \left(\eta_w - \frac{\eta_w^2 \tilde{M}}{2}\right)\|g_w^t\|^2 + \frac{\eta_w^2 \tilde{M}\sigma_w^2}{2} \tag{67}$$

Rearranging the terms, we obtain

$$\left(\eta_w - \frac{\eta_w^2 \tilde{M}}{2}\right)\|g_w^t\|^2 \leq \mathbb{E}\left[\sum_{i=1}^n p_i^t \hat{F}_i(w^t) - \sum_{i=1}^n p_i^t \hat{F}_i(w^{t+1})|\mathcal{F}^t\right] + \frac{\eta_w^2 \tilde{M}\sigma_w^2}{2} \tag{68}$$

$$= \mathbb{E}\left[\sum_{i=1}^n p_i^t \hat{F}_i(w^t) - \sum_{i=1}^n p_i^{t+1} \hat{F}_i(w^{t+1})|\mathcal{F}^t\right] \tag{69}$$

$$+ \mathbb{E}\left[\sum_{i=1}^n p_i^{t+1} \hat{F}_i(w^{t+1}) - \sum_{i=1}^n p_i^t \hat{F}_i(w^{t+1})|\mathcal{F}^t\right] + \frac{\eta_w^2 \tilde{M}\sigma_w^2}{2} \tag{70}$$

We bound the second expectation in the above equation:

$$\mathbb{E}\left[\sum_{i=1}^n p_i^{t+1} \hat{F}_i(w^{t+1}) - \sum_{i=1}^n p_i^t \hat{F}_i(w^{t+1})|\mathcal{F}^t\right] = \mathbb{E}\left[\sum_{i=1}^n (p_i^{t+1} - p_i^t)\hat{F}_i(w^{t+1})|\mathcal{F}^t\right]$$

$$\leq \mathbb{E}\left[\|p^{t+1} - p^t\|_2 \sum_{i=1}^n \left(\hat{F}_i(w^{t+1})\right)^{1/2}|\mathcal{F}^t\right] \tag{71}$$

$$\leq \sqrt{n}\hat{B}\mathbb{E}\left[\|p^{t+1} - p^t\|_2|\mathcal{F}^t\right] \tag{72}$$

$$\leq 2\sqrt{n}\hat{B}(\mathbb{E}\left[\|\eta_p \hat{g}_p^t\|_2|\mathcal{F}^t\right]) \tag{73}$$

$$= 2\eta_p \sqrt{n}\hat{B}\hat{G}_p \tag{74}$$

where (71) follows from the Cauchy-Shwarz Inequality, (72) follows by the bound on $\hat{f}_{i,j}$ for all $i$, (73) follows from the update rule for $p$ combined with the projection property (since $p^t \in \Delta_n$, $\|p^t - (p^t + \eta_p \hat{g}_p^t)\| \geq \|p^t - \Pi_{\Delta_n}(p^t + \eta_p \hat{g}_p^t)\|$), and (74) follows by Lemma 1, noting $\hat{G}_p^2 = \frac{n(n+C-1)\hat{B}}{C}$. Using this result, summing (70) from $t = 1$ to $T$, and taking the expectation over all the stochastic gradients of both sides and using the Law of Iterated Expectations to remove the conditioning on $\mathcal{F}^t$, we obtain

$$\left(\eta_w - \frac{\eta_w^2 \tilde{M}}{2}\right)\sum_{t=1}^T \mathbb{E}\left[\|g_w^t\|^2\right]$$

$$\leq \mathbb{E}\left[\sum_{i=1}^n p_i^1 \hat{F}_i(w^1)\right] - \mathbb{E}\left[\sum_{i=1}^n p_i^{T+1} \hat{F}_i(w^{T+1})\right] + 2T\eta_p \sqrt{n}\hat{B}\hat{G}_p + \frac{T\tilde{M}\eta_w^2\sigma_w^2}{2}$$

$$\leq \phi(w^1, p^1) + \hat{B} + 2T\eta_p \sqrt{n}\hat{B}\hat{G}_p + \frac{T\eta_w^2 \tilde{M}\sigma_w^2}{2}$$

Next, dividing both sides by $T\left(\eta_w - \frac{\eta_w^2 \tilde{M}}{2}\right)$ we have

$$\frac{1}{T}\sum_{t=1}^T \mathbb{E}\left[\|g_w^t\|^2\right] \leq \frac{\phi(w^1, p^1) + \hat{B}}{T\left(\eta_w - \frac{\eta_w^2 \tilde{M}}{2}\right)} + \frac{2\eta_p \sqrt{n}\hat{B}\hat{G}_p}{\left(\eta_w - \frac{\eta_w^2 \tilde{M}}{2}\right)} + \frac{\eta_w \tilde{M}\sigma_w^2}{\left(2 - \eta_w \tilde{M}\right)} \tag{75}$$

which by (62) is the desired bound on $\mathbb{E}[\|\nabla_w \phi(w_T^\tau, p_T^\tau)\|_2^2]$.

Next we show the bound on the optimality of $p_T^\tau$. As before, we start by evaluating the expectation over $\tau$:

$$\mathbb{E}\left[\phi(w_T^\tau, p_T^\tau)\right] = \mathbb{E}\left[\mathbb{E}_\tau\left[\phi(w_T^\tau, p_T^\tau)\right]\right] \tag{76}$$

$$= \mathbb{E}\left[\frac{1}{T}\sum_{t=1}^T \phi(w^t, p^t)\right] \tag{77}$$

$$= \frac{1}{T}\sum_{t=1}^T \mathbb{E}\left[\phi(w_T^\tau, p_T^\tau)\right] \tag{78}$$

Next, since $\phi(w, p)$ is linear in $p$, we have that for any $p \in \Delta_n$ and any $t \in \{1, ..., T\}$,

$$\mathbb{E}\left[\phi(w^t, p) - \phi(w^t, p^t)|\mathcal{F}^t\right] = \mathbb{E}\left[(p - p^t)g_p^t|\mathcal{F}^t\right]$$
$$= \mathbb{E}\left[(p - p^t)\hat{g}_p^t|\mathcal{F}^t\right] + \mathbb{E}\left[(p - p^t)(g_p^t - \hat{g}_p^t)|\mathcal{F}^t\right] \tag{79}$$
$$= \mathbb{E}\left[(p - p^t)\hat{g}_p^t|\mathcal{F}^t\right] \tag{80}$$

where (80) follows because $\hat{g}_p^t$ is an unbiased estimate of $g_p^t$. Using (80) and the identity $2ab = a^2 + b^2 - (a - b)^2$ with $a = p - p^t$ and $b = \eta_p \hat{g}_p^t$ yields

$$\mathbb{E}\left[\phi(w^t, p) - \phi(w^t, p^t)|\mathcal{F}^t\right] = \mathbb{E}\left[\frac{1}{2\eta_p}\left(\|p - p^t\|_2^2 + (\eta_p)^2\|\hat{g}_p^t\|_2^2 - \|p - (p^t + \eta_p\hat{g}_p^t)\|_2^2\right)|\mathcal{F}^t\right] \tag{81}$$

$$\leq \mathbb{E}\left[\frac{1}{2\eta_p}\left(\|p - p^t\|_2^2 + (\eta_p)^2\|\hat{g}_p^t\|_2^2 - \|p - p^{t+1}\|_2^2\right)|\mathcal{F}^t\right] \tag{82}$$

$$\leq \mathbb{E}\left[\frac{1}{2\eta_p}\left(\|p - p^t\|_2^2 + (\eta_p)^2\hat{G}_p^2 - \|p - p^{t+1}\|_2^2\right)|\mathcal{F}^t\right] \tag{83}$$

where (82) follows from the projection property and (83) follows from Lemma 1. Summing from $t = 1$ to $T$ and taking the expectation over all the stochastic gradients of both sides and using the Law of Iterated Expectations to remove the conditioning on $\mathcal{F}^t$, we obtain

$$\sum_{t=1}^T \mathbb{E}\left[\phi(w^t, p) - \phi(w^t, p^t)\right] \leq \sum_{t=1}^T \frac{1}{2\eta_p}\mathbb{E}\left[\|p - p^t\|_2^2\right] - \frac{1}{2\eta_p}\mathbb{E}\left[\|p - p^{t+1}\|_2^2\right] + \frac{\eta_p}{2}\hat{G}_p^2 \tag{84}$$

$$= \frac{1}{2\eta_p}\mathbb{E}\left[\|p - p^1\|_2^2\right] + \frac{\eta_p}{2}T\hat{G}_p^2 \tag{85}$$

$$\leq \frac{1}{\eta_p} + \frac{\eta_p T\hat{G}_p^2}{2} \tag{86}$$

where (84) follows from the telescoping sum and (86) follows from the fact that $p, p^1 \in \Delta_n$ and $\Delta_n$ is contained in an $\ell_2$ ball of radius 1. Dividing both sides of (86) by $T$ and rearranging terms

$$\frac{1}{T}\sum_{t=1}^T \mathbb{E}\left[\phi(w^t, p^t)\right] \geq \mathbb{E}\left[\phi(w_T^\tau, p)\right] - \left(\frac{1}{\eta_p T} + \frac{\eta_p \hat{G}_p^2}{2}\right) \tag{87}$$

Finally, since (87) holds for all $p \in \Delta_n$, we maximize the right hand side over $p \in \Delta_n$, yielding

$$\frac{1}{T}\sum_{t=1}^T \mathbb{E}\left[\phi(w^t, p^t)\right] \geq \max_{p \in \Delta_n}\left[\phi(w_T^\tau, p)\right] - \left(\frac{1}{\eta_p T} + \frac{\eta_p \hat{G}_p^2}{2}\right)$$

From (78), the left hand side above is equal to $\mathbb{E}\left[\phi(w_T^\tau, p_T^\tau)\right]$, thus completing the proof. $\qquad\square$

Theorem 2 follows immediately from Proposition 2 by setting the step sizes appropriately.

# H   Proof of Theorem 3

First we have the following proposition for unspecified constant stepsizes.

**Proposition 3.** *Suppose Assumptions 1 and 2 hold and $\mathcal{W}$ is convex and compact. Let the step sizes $\eta_w^t$ and $\eta_p^t$ be constant over all $t$, denoted by $\eta_w$ and $\eta_p$, respectively, where $\eta_w < (2/\tilde{M})$. Let $(w_T^\tau, p_T^\tau)$ be the solution returned by Algorithm 1 after $T$ iterations. Then we have*

$$\mathbb{E}[\|\bar{g}_w(w_T^\tau, p_T^\tau)\|_2^2] \leq \frac{2(\phi(w^1, p^1) + \hat{B})}{T(2\eta_w - \eta_w^2 \tilde{M})} + \frac{4\eta_p \sqrt{n} \hat{B} \hat{G}_p}{(2\eta_w - \eta_w^2 \tilde{M})} + \frac{\sigma_w^2}{(2 - \eta_w \tilde{M})},$$

$$\mathbb{E}[\phi(w_T^\tau, p_T^\tau)] \geq \max_{p \in \Delta_n} \{\mathbb{E}[\phi(w_T^\tau, p)]\} - \frac{1}{\eta_p T} - \frac{\eta_p \hat{G}_p^2}{2}.$$

*Proof.* Here it is helpful to rewrite $\Pi_\mathcal{W}$ as a prox operation. Defining $I_\mathcal{W} : \mathcal{W} \to \{0, +\infty\}$ as $I_\mathcal{W}(w) = 0$ if $w \in \mathcal{W}$ and $I_\mathcal{W}(w) = +\infty$ otherwise, the update rule for $w$ becomes:

$$w^{t+1} = \Pi_\mathcal{W}(w^t - \eta_w^t \hat{g}_w^t) = \underset{u \in \mathbb{R}^d}{\mathrm{argmin}}\{\langle \hat{g}_w^t, u \rangle + \frac{1}{2\eta_w^t}\|u - w^t\|_2^2 + I_\mathcal{W}(u)\} = \mathrm{prox}_{\eta_w^t I_\mathcal{W}}(w^t - \eta_w^t \hat{g}_w^t)$$

and the projected stochastic gradient is equivalent to

$$\bar{g}_w^t = \frac{1}{\eta_w^t}(w^t - \mathrm{prox}_{\eta_w^t I_\mathcal{W}}(w^t - \eta_w^t \hat{g}_w^t))$$

The rewritten objective, using $I_\mathcal{W}$ to remove the constraint on $w$, is as follows:

$$\min_{w \in \mathbb{R}^d} \max_{p \in \Delta_n} \{\Phi(w, p) := \phi(w, p) + I_\mathcal{W}(w)\} \tag{88}$$

With these notations in hand, we are ready to begin the proof. We make analogous initial arguments to those in the proof of Theorem 2 in [14], and cite two results on the properties of the prox operation from the same paper. By the $\tilde{M}$-smoothness of $\hat{F}_i$ for each $i$, we have equation (63), and thus for any $t \in \{1, ..., T\}$,

$$\sum_{i=1}^n p_i^t \hat{F}_i(w^{t+1}) \leq \sum_{i=1}^n p_i^t \hat{F}_i(w^t) + \left(\nabla_w \sum_{i=1}^n p_i^t \hat{F}_i(w^t)\right)^T (w^{t+1} - w^t) + \frac{\tilde{M}}{2}\|w^{t+1} - w^t\|_2^2$$

$$= \sum_{i=1}^n p_i^t \hat{F}_i(w^t) - \eta_w^t \left(\nabla_w \sum_{i=1}^n p_i^t \hat{F}_i(w^t)\right)^T \bar{g}_w^t + \frac{\tilde{M}}{2}(\eta_w^t)^2\|\bar{g}_w^t\|_2^2$$

$$= \sum_{i=1}^n p_i^t \hat{F}_i(w^t) - \eta_w^t (\hat{g}_w^t)^T \bar{g}_w^t + \frac{\tilde{M}}{2}(\eta_w^t)^2\|\bar{g}_w^t\|_2^2 + \eta_w^t (\delta_w^t)^T \bar{g}_w^t$$

where in the identity we have used the definitions of $\bar{g}^t$ and $\delta_w^t$. Next, using Lemma 1 in [14] with $x = w^t$, $\gamma = \eta_w^t$, and $g = \hat{g}_w^t$, we obtain

$$\sum_{i=1}^n p_i^t \hat{F}_i(w^{t+1}) \leq \sum_{i=1}^n p_i^t \hat{F}_i(w^t) - [\eta_w^t \|\bar{g}^t\|_2^2 + I_\mathcal{W}(w^{t+1}) - I_\mathcal{W}(w^t)] + \frac{\tilde{M}}{2}(\eta_w^t)^2\|\bar{g}_w^t\|_2^2 + \eta_w^t (\delta_w^t)^T \bar{g}_w^t$$

$$= \sum_{i=1}^n p_i^t \hat{F}_i(w^t) - [\eta_w^t \|\bar{g}^t\|_2^2 + I_\mathcal{W}(w^{t+1}) - I_\mathcal{W}(w^t)] + \frac{\tilde{M}}{2}(\eta_w^t)^2\|\bar{g}_w^t\|_2^2$$

$$+ \eta_w^t (\delta_w^t)^T g^t + \eta_w^t (\delta_w^t)^T (\bar{g}_w^t - g^t)$$

where $\delta_w^t := \hat{g}_w^t - g_w^t$ and $g^t := \frac{1}{\eta_w^t}(w^t - \mathrm{prox}_{\eta_w^t I_\mathcal{W}}(w^t - \eta_w^t g_w^t))$ is the projected full gradient with respect to $w$. Thus after rearranging terms,

$$\Phi(w^{t+1}, p^t) \leq \Phi(w^t, p^t) - \left(\eta_w^t - \frac{\tilde{M}}{2}(\eta_w^t)^2\right)\|\bar{g}_w^t\|_2^2 + \eta_w^t \langle \delta_w^t, g^t \rangle + \eta_w^t \|\delta_w^t\|\|\bar{g}_w^t - g^t\|$$

$$\leq \Phi(w^t, p^t) - \left(\eta_w^t - \frac{\tilde{M}}{2}(\eta_w^t)^2\right)\|\bar{g}_w^t\|_2^2 + \eta_w^t \langle \delta_w^t, g^t \rangle + \eta_w^t \|\delta_w^t\|^2$$

where the last inequality follows from Proposition 1 in [14] with $x = w^t$, $\gamma = \eta_w^t$, $g_1 = \hat{g}_w^t$, and $g_2 = g_w^t$. Rearranging terms, we have

$$\left(\eta_w^t - \frac{\tilde{M}}{2}(\eta_w^t)^2\right)\|\bar{g}_w^t\|_2^2$$

$$\leq \Phi(w^t, p^t) - \Phi(w^{t+1}, p^t) + \eta_w^t\langle\delta_w^t, g^t\rangle + \eta_w^t\|\delta_w^t\|^2$$
$$= \left(\Phi(w^t, p^t) - \Phi(w^{t+1}, p^{t+1})\right) + \left(\Phi(w^{t+1}, p^{t+1}) - \Phi(w^{t+1}, p^t)\right) + \eta_w^t\langle\delta_w^t, g^t\rangle + \eta_w^t\|\delta_w^t\|^2$$
$$= \left(\Phi(w^t, p^t) - \Phi(w^{t+1}, p^{t+1})\right) + \left(\phi(w^{t+1}, p^{t+1}) - \phi(w^{t+1}, p^t)\right) + \eta_w^t\langle\delta_w^t, g^t\rangle + \eta_w^t\|\delta_w^t\|^2$$

Taking the expectation with respect to the stochastic gradients conditioned on the history up to time $t$ of each side, we have

$$\left(\eta_w^t - \frac{\tilde{M}}{2}(\eta_w^t)^2\right)\mathbb{E}\left[\|\bar{g}_w^t\|_2^2|\mathcal{F}^t\right]$$

$$\leq \mathbb{E}\left[\left(\Phi(w^t, p^t) - \Phi(w^{t+1}, p^{t+1})\right)|\mathcal{F}^t\right] + \mathbb{E}\left[\left(\phi(w^{t+1}, p^{t+1}) - \phi(w^{t+1}, p^t)\right)|\mathcal{F}^t\right]$$
$$+ \eta_w^t\mathbb{E}\left[\langle\delta_w^t, g^t\rangle|\mathcal{F}^t\right] + \eta_w^t\mathbb{E}\left[\|\delta_w^t\|^2|\mathcal{F}^t\right]$$
$$= \mathbb{E}\left[\left(\Phi(w^t, p^t) - \Phi(w^{t+1}, p^{t+1})\right)|\mathcal{F}^t\right] + \mathbb{E}\left[\sum_{i=1}^n (p_i^{t+1} - p_i^t)\hat{F}_i(w^{t+1})|\mathcal{F}^t\right]$$
$$+ \eta_w^t\mathbb{E}\left[\langle\delta_w^t, g^t\rangle|\mathcal{F}^t\right] + \eta_w^t\mathbb{E}\left[\|\delta_w^t\|^2|\mathcal{F}^t\right] \tag{89}$$

Note that we can use the Holder Inequality to bound the second expectation in (89). In doing so we obtain

$$\left(\eta_w^t - \frac{\tilde{M}}{2}(\eta_w^t)^2\right)\mathbb{E}\left[\|\bar{g}_w^t\|_2^2|\mathcal{F}^t\right]$$

$$\leq \mathbb{E}\left[\left(\Phi(w^t, p^t) - \Phi(w^{t+1}, p^{t+1})\right)|\mathcal{F}^t\right] + \mathbb{E}\left[\|p^{t+1} - p^t\|_2\left(\sum_{i=1}^n \hat{F}_i(w^{t+1})^2\right)^{1/2}|\mathcal{F}^t\right]$$

$$+ \eta_w^t\mathbb{E}\left[\langle\delta_w^t, g^t\rangle|\mathcal{F}^t\right] + \eta_w^t\mathbb{E}\left[\|\delta_w^t\|^2|\mathcal{F}^t\right]$$
$$\leq \mathbb{E}\left[\left(\Phi(w^t, p^t) - \Phi(w^{t+1}, p^{t+1})\right)|\mathcal{F}^t\right] + 2\sqrt{n}B\mathbb{E}\left[\|\eta_p^t\hat{g}_p^t\|_2|\mathcal{F}^t\right] + \eta_w^t\mathbb{E}\left[\langle\delta_w^t, g^t\rangle|\mathcal{F}^t\right]$$
$$+ \eta_w^t\mathbb{E}\left[\|\delta_w^t\|_2^2|\mathcal{F}^t\right] \tag{90}$$
$$\leq \mathbb{E}\left[\left(\Phi(w^t, p^t) - \Phi(w^{t+1}, p^{t+1})\right)|\mathcal{F}^t\right] + 2\sqrt{n}B\eta_p^t\hat{G}_p + \eta_w^t\mathbb{E}\left[\langle\delta_w^t, g^t\rangle|\mathcal{F}^t\right] + \eta_w^t\mathbb{E}\left[\|\delta_w^t\|_2^2|\mathcal{F}^t\right] \tag{91}$$

$$\leq \mathbb{E}\left[\left(\Phi(w^t, p^t) - \Phi(w^{t+1}, p^{t+1})\right)|\mathcal{F}^t\right] + 2\sqrt{n}B\eta_p^t\hat{G}_p + \eta_w^t\mathbb{E}\left[\|\delta_w^t\|_2^2|\mathcal{F}^t\right] \tag{92}$$
$$\leq \mathbb{E}\left[\left(\Phi(w^t, p^t) - \Phi(w^{t+1}, p^{t+1})\right)|\mathcal{F}^t\right] + 2\sqrt{n}B\eta_p^t\hat{G}_p + \eta_w^t\sigma_w^2 \tag{93}$$

where (90) follows from the definition of $B$ and the update rule for $p$ combined with the projection property, (91) follows from the definition of $\hat{G}_p$, (92) follows from the facts that $g^t$ is a deterministic function of the stochastic samples that determine the stochastic gradients up to time $t$ and $\hat{g}_w^t$ is an unbiased estimate of $g_w^t$, and (93) follows from the computation of $\mathbb{E}[\|\delta_w\|^2]$ given in Lemma 4. Summing over $t = 1, ..., T$, setting the step sizes to be constants, and taking the expectation with respect to all of the stochastic gradients and using the Law of Iterated Expectations, we find

$$\left(\eta_w - \frac{\tilde{M}}{2}(\eta_w)^2\right)\sum_{t=1}^T \mathbb{E}\left[\|\bar{g}_w^t\|_2^2\right] \leq \Phi(w^1, p^1) - \mathbb{E}\left[\Phi(w^{T+1}, p^{T+1})\right] + 2T\eta_p B\sqrt{n}\hat{G}_p + T\eta_w\sigma_w^2$$

$$\leq \Phi(w^1, p^1) + B + 2T\eta_p B\sqrt{n}\hat{G}_p + T\eta_w\sigma_w^2$$

Next we divide both sides by $T\left(\eta_w - \frac{\tilde{M}}{2}(\eta_w)^2\right)$ to yield

$$\frac{1}{T}\sum_{t=1}^T \mathbb{E}\left[\|\bar{g}_w^t\|_2^2\right] \leq \frac{2(\phi(w^1, p^1) + B)}{T(2\eta_w - \eta_w^2\tilde{M})} + \frac{4\eta_p\sqrt{n}B\hat{G}_p}{(2\eta_w - \eta_w^2\tilde{M})} + \frac{\sigma_w^2}{(2 - \eta_w\tilde{M})}$$

Using an analogous argument as (62), we have that the left hand side of the above equation is equal to $\mathbb{E}[\|\bar{g}_w(w_T^\tau, p_T^\tau)\|_2^2]$, thus we have completed the proof of the convergence result in $w$.

For the convergence with respect to $p$, note that the update rule for $p^{t+1}$ is identical to the update rule analyzed in Proposition 2, and the output procedure is the same for both algorithms. Furthermore, since the convergence analysis of $p$ does not depend on the update rule for $w$, the analysis with respect to $p$ in the proof of Proposition 2 still applies here, thus we have the same bound. □

The only significant difference between the bound in Proposition 3 and the bound derived in Proposition 2 is that the term with $\sigma_w^2$ is not multiplied by the step size $\eta_w$, thus appears to asymptotically behave as a constant. Therefore, in order to show that the right hand side in the above bound converges, we must treat $\sigma_w^2$ as a function of the number of stochastic gradients computed during each iteration. Recall that $\sigma_w^2$ is an upper bound on $\mathbb{E}\|\hat{g}_w - g_w^t\|_2^2$, and note from Lemma 4 that we can write it as $\sigma_w^2 = \tilde{\sigma}_w^2/C$, where $\tilde{\sigma}_w^2$ does not depend on $C$ or $T$, and $C$ is the number of sampled task instances used for each stochastic gradient computation, and each sampled task instance involves a constant number of function, gradient and Hessian evaluations. We can therefore define $C$ as an increasing function of $T$ in order for $\sigma_w^2$ to decrease with $T$, while the total number of oracle evaluations performed by the algorithm will be $\mathcal{O}(CT)$.

To balance terms, we must choose $\eta_p$ and $\sigma_w^2$ to be of the same order with respect to $T$. Thus for some $\beta \in (0,1)$, let $\eta_p = \mathcal{O}(T^{-\beta})$ and $C = \mathcal{O}(T^\beta)$. Since here $C$ grows with $T$, we can assume without loss of generality that $C > n$ (since if this were not the case, the only way we would get improvement over the 1/5 rate, to 1/4, would require $\beta = 1$, which would mean C = m = n = T, which is not realistic). In this case, $\hat{G}_p^2$ can be numerically upper bounded as

$$\hat{G}_p^2 := \frac{n(n+C-1)}{C}\hat{B}^2 = (\frac{n^2}{C} + n - \frac{n}{C})\hat{B}^2 \le 2n\hat{B}^2 \tag{94}$$

Replacing $\hat{G}_p^2$ with this upper bound in the results from Proposition 3 and plugging in the appropriate step sizes completes the proof of Theorem 3.

# I  Generalization Results

## I.1  Proof of Proposition 1

The result is a standard Rademacher complexity bound, see for example [23], thus we omit the proof.

## I.2  Proof of Theorem 4

*Proof.* Since $\mathcal{D}_{n+1}^K \times \mathcal{D}_{n+1}^J$ is a mixture distribution, we have, for any $w$,

$$F_{n+1}(w) = \mathbb{E}_{(D_{n+1,j}^{\text{train}}, D_{n+1,j}^{\text{test}}) \sim \mathcal{D}_{n+1}}[\hat{f}_{n+1,j}(w - \alpha\nabla\hat{f}_{n+1,j}(w, D_{n+1,j}^{\text{train}}), D_{n+1,j}^{\text{test}})] \tag{95}$$

$$= \sum_{i=1}^n a_i \mathbb{E}_{(D_{n+1,j}^{\text{train}}, D_{n+1,j}^{\text{test}}) \sim \mathcal{D}_i}[\hat{f}_{n+1,j}(w - \alpha\nabla\hat{f}_{n+1,j}(w, D_{n+1,j}^{\text{train}}), D_{n+1,j}^{\text{test}})] \tag{96}$$

$$= \sum_{i=1}^n a_i F_i(w) \tag{97}$$

Therefore, using Proposition 1 and a union bound over the $n$ tasks, we have that with probability at least $1 - n\delta'$ over the choice of samples used to compute $\hat{F}_i(w)$,

$$F_{n+1}(w^*) = \sum_{i=1}^n a_i F_i(w^*) \le \sum_{i=1}^n a_i \hat{F}_i(w^*) + 2a_i \mathfrak{R}_{m_i}^i(\mathcal{F}) + a_i \hat{B}\sqrt{\frac{\log 1/\delta'}{2m_i}} \tag{98}$$

Table 4: Omniglot $N$-way, $K$-shot classification accuracies (%). After meta-training, 5,000 few-shot classification problems (task instances) are sampled uniformly from the 25 alphabets (tasks) used for meta-training, likewise for the 20 new meta-testing alphabets. For each alphabet, the average accuracy on task instances from that alphabet is computed, and statistics are taken across these average accuracies. 'Weighted Mean' weighs the alphabet accuracies by the meta-training distribution, which corresponds to the quantity MAML aims to optimize, whereas 'Mean' weighs all alphabets equally. 'Worst' is the minimum alphabet accuracy, and 'Std. Dev.' is the standard deviation across the alphabet accuracies, with 95% confidence intervals given over three full runs for all statistics.

| | | Meta-training Alphabets | | | Meta-testing Alphabets | | |
|---|---|---|---|---|---|---|---|
| $(N, K)$ | Algorithm | Weighted Mean | Mean | Worst | Mean | Worst | Std. Dev. |
| (10,1) | MAML | $\mathbf{98.5 \pm 1.2}$ | $91.0 \pm .4$ | $54.5 \pm 2.5$ | $\mathbf{85.9 \pm 0.3}$ | $\mathbf{71.0 \pm 1.2}$ | $\mathbf{6.3 \pm .1}$ |
| | TR-MAML | $95.6 \pm .3$ | $\mathbf{94.0 \pm .1}$ | $\mathbf{89.5 \pm 1.0}$ | $83.6 \pm .5$ | $70.2 \pm 2.4$ | $6.6 \pm .3$ |
| (10,5) | MAML | $\mathbf{99.1 \pm .1}$ | $95.0 \pm .1$ | $70.1 \pm 2.8$ | $92.1 \pm .1$ | $82.9 \pm 0.1$ | $3.8 \pm .1$ |
| | TR-MAML | $98.5 \pm .4$ | $\mathbf{98.6 \pm .4}$ | $\mathbf{96.2 \pm 1.0}$ | $\mathbf{93.8 \pm .7}$ | $\mathbf{87.7 \pm 1.4}$ | $\mathbf{3.2 \pm .5}$ |

Making the substitution $\delta = n\delta'$ and using the fact that $a_i \in \Delta_n$ and the definition of $w^*$ yields that

$$F_{n+1}(w^*) \leq \max_{p \in \Delta_n} \sum_{i=1}^{n} p_i \hat{F}_i(w^*) + 2a_i \mathfrak{R}_{m_i}^i(\mathcal{F}) + a_i \hat{B} \sqrt{\frac{\log(n/\delta)}{2m_i}} \tag{99}$$

$$= \min_{w \in \mathcal{W}} \max_{p \in \Delta_n} \sum_{i=1}^{n} p_i \hat{F}_i(w^*) + 2a_i \mathfrak{R}_{m_i}^i(\mathcal{F}) + a_i \hat{B} \sqrt{\frac{\log(n/\delta)}{2m_i}} \tag{100}$$

$$\tag{101}$$

with probability at least $1 - \delta$, which completes the proof. $\qquad\square$

# J   Additional Experimental Results and Details

We performed all experiments on a 3.7GHz, 6-core Intel Corp i7-8700K CPU. For all experiments, there was no significant difference in the time required to run TR-MAML compared to MAML.

## J.1   Sinusoid Regression

For the sinusoid regression experiments, we adapted the codebase from the original MAML paper [12] available at `https://github.com/cbfinn/maml`, which is written in in Tensorflow `https://www.tensorflow.org/`. We used a batch size of 25 task instances with $J$ (the number of evaluation points in each task instance/few-shot learning episode) equal to $K$. We set $\eta_w = 10^{-3}$, $\alpha = 10^{-3}$, and used one step of SGD update and the Adam optimizer for the meta-learning update step for $w$ for both TR-MAML and MAML, consistent with the original sinusoid experiments [12]. To update $p$ in TR-MAML, we used vanilla projected SGD (without an optimizer) with learning rate $\eta_p = 0.0001$ when $K = 5$ and $\eta_p = 0.0002$ when $K = 10$.

## J.2   Few-shot Image Classification

For the image classification experiments, we adapted the codebase from the repository available at `https://github.com/AntreasAntoniou/HowToTrainYourMAMLPytorch` that implements in Pytorch `https://pytorch.org/` the experiments in the paper [1]. Again we kept most of the default parameters consistent. The Adam optimizer was used for the meta-update of $w$ and vanilla SGD was used to update $p$. We set $\eta_p = 2.0 \times 10^{-5}$ for the 5-way experiments, $\eta_p = 1.6 \times 10^{-5}$ for the 10-way experiments, and $\eta_p = 1.0 \times 10^{-5}$ for the 20-way experiments. In all cases, we set $J = 10$. After meta-training for 60,000 iterations with a batch size of 8, the most recent meta-trained model was evaluated on both the meta-testing and meta-training tasks (alphabets). One step of SGD was used for both meta-training and meta-testing in all experiments. Images were augmented by rotations of 90 degrees, with augmented images considered part of the same class (thus there were $20 \times 4 = 80$ images per class), but each image in each class in each task instance was rotated by the same amount. Additional results for the 10-way classification case are shown in Table 4.

For the *mini*-ImageNet experiments, we set $\eta_p = 1.6 \times 10^{-5}$ and $J = 15$, and execute for 60,000 iterations with a batch size of 2 task instances. We use 5 steps of inner gradient updates during both meta-training and meta-testing.