[Reviews · NeurIPS 2020]

Review 1

Summary and Contributions: The submission proposes a modification of the multi-task meta-learning objective from the average of the per-task losses to the maximum of those losses. The argument is that this will force the learner to learn all tasks to a comparable amount, even the worst-case ones, so no task can be ignored. The manuscript presents a stochastic algorithm for optimizing the resulting min-max problem and proves convergence including explicit rates for the convex as well as the non-convex case. Some generalization bounds are also stated. Experiments on small-scale data show that worst-case (w.r.t. tasks) performance is improved compared to MAML, which optimizes for average loss.

Strengths: + meta-learning is a problem of high relevance and of high interest in the community at the moment + the manuscript contains a lot of material + the proposed methods seems technically correct + the proposed algorithm is not only tested empirically, but also its convergence is formally proved, including rates + the experiments indeed show that performance on the worst tasks are improved, in the Omniglot case, even average performance sometimes gets better

Weaknesses: The work does not seem to have any major flaws, but there's a bunch of small to medium-sized weaknesses regarding the novelty, significance and relevance of the work. 1) the actual new objective is not of high novelty: switching from average to max is a well-known way of aiming for better worst-case performance 2) the motivation/discussion why to optimize for worst-case task loss is insufficient (see below) 3) the contribution on the side of optimization is not made clear enough. The proposed algorithm and its convergence analysis are based on prior work for min-max problems (of course), and it is not explained in how far the proposed steps are simply an application of that. 4) the established rates for the non -convex case ( max(1/eps^5,, 1/delta^5) ) are far from practical 5) the the generalization bounds are not useful as given, because they rely on a task-specific notion of Rademacher complexity, that is not explained, and not quantified. 6) the manuscript has no conclusion section, page 8 ends on experiments (the "Broader Impact" Section on page 9 is written as an Conclusion, but I disregard this, as that is not that section is meant to be, and it is beyond the page limit) 7) experiments are mostly unsurprising: switching from average to max loss, the average-case performance generally gets worse, the worst-case performance gets better Details: 2) The manuscript's arguments in favor of minimizing the max-loss across tasks is not fully convincing. The proposed objective is not "robust" as suggested in the manuscript's title, but it is brittle. A single "outlier" or "too hard" task would render the setting pointless. The manuscript mentions that adversarial tasks would be a problem, but an adversary is not required, already tasks of different difficulty (e.g. different Bayes error rates) should pose problems. Prior work (which is cited, e.g. [32],[9]) states explicitly that avg-loss has a lot of advantage, but that there is situations in which minimizing the max-loss can make sense. That, however, is across samples which come from the same distribution and emphasis is on the realizable setting, i.e. even the max can be 0, and the different to average loss is mainly of the optimization side. For multiple tasks, the differences and problem seem far bigger. I would have hoped to see a discussion of this, and potentially a justification. 5) The generalization bounds rely on standard arguments, which use the task-specific Rademacher complexity as a black box. For the reader to understand the implications of the bounds, the reader has to understand the behavior of the Rademacher complexity. Does it even converge to 0 for m->infty? Is the amount of test or the train data crucial for that? What's the dependence on \mathcal{W}? Maybe it could be expressed in terms of other existing, better understood, Rademacher complixity measures?

Correctness: I did not spot any mistakes. I did not check the proofs in the supplemental material, though.

Clarity: The writing is not ideal. Overall, the paper is trying to squeeze too much into the available pages. The work feels almost like three papers: one that presents an algorithmic with its convergence analysis, one that presents an objective and some generalization bounds, and one that show experimental improvements. Ultimately, each part ended up a bit too shor to be satisfying. - the motivation of the max-loss is not convincing. I don't know if this is fixable, but making clearer in which situation is it a good a idea and when it is not might help. - the use of 'task instances' and 'episodes' in the Problem Formulation (lines 93-98) is not clear enough. I only inferred what is meant from the later text and the treatment of the j-index. - the algorithm and convergence analysis lacks a clear distinction of what is standard techniques/results and what is a new contribution. - the generalization bounds are rather useless to the reader, because the properties of the occurring version of Rademacher complexity is not explained. The results for convex combinations of tasks is unsurprising given the max-formulation, but convex combination of tasks are not a very realistic setting anyway. - the experiments do not convince me that the result carry over to "real-world" tasks. Results on two datasets are reported, but both are quite artificial (sinusoid regression is synthetic 1D, Omniglot is handwritten characters). - the manuscript has no conclusion section, page 8 ends on experiments (the "Broader Impact" Section on page 9 is written as an Conclusion, but I have to disregard this, as that is not that section is meant to be, and it is beyond the page limit)

Relation to Prior Work: Prior works is acknowledged properly. The exact differences of the proposed steps to the ones from the literature are not clear enough, though.

Reproducibility: No

Additional Feedback: Dear authors, I appreciate the amount of material presented, but the density of writing in the manuscript makes it hard for me as reader to assess the aspects of novelty, relevance and significance. That's why I gave a borderline score, but I'd be happy to still adjust that. To make me better understand the contribution of the paper, could you please explicitly list what exactly you consider your main contributions (ideally split into each of the parts: setting, algorithm, convergence analysis, bounds, experiments)? Specifically for the algorithm, convergence analysis and bounds I would be interested which parts you consider applications of existing work (with might adjustments), and which ones you consider as new contributions to the community? ----------------------- After reading the reviewer response and following the discussion, my impression of the work is still the same. It makes some contribution, which are laid out in the rebuttal (thank you), but none of them appear a fundamentally new contribution. The motivation for max-loss is unconvincing to me, unless make stronger assumptions on the task environment. Overall, I remain with the assessment that this is a borderline work. Comparable work has been published at NeurIPS, and comparable work has been rejected.


Review 2

Summary and Contributions: This paper proposes a new meta-learning objective to learn rare tasks on equal importance with major tasks. It causes robust learning to distribution shifts on the observed tasks and better generalization performance than the average loss. The authors also prove that their formulation convergences in both convex and nonconvex settings, and show an empirical gain on regression and image classification experiments.

Strengths: The task-robust (robust learning not just for significant tasks but also rare or hard tasks) is a nice idea to prevent task overfitting (given tasks can be biased when the number of it is small) or make a more general meta-learned model, and they provided the prove about the convergences of their objective.

Weaknesses: To use the proposed objective, we need to make the task. When the task distribution is continuous, we need to quantize the distribution. How we quantize, it can be another hyperparameter to tune. Another that I want to mention is that it can cause worse performance on an easy task. Let me give an example. When there are easy and difficult tasks, the proposed objective focuses on difficult tasks. During training, the loss for the easy task can be larger than hard. At that time, the model tries to learn more about the easy task. However, the performance on the easy task cannot be much better than hard, because if it happens, the objective will focus on the hard task more. Thus, the model can do better on the easy task, but it cannot due to the objective. (I think the lower MEAN performance on regression than MAML is for this reason.) The limitation of this paper is lack of empirical analysis on more complicated image classification task (mini-imagenet) or RL tasks (point navigation or tasks on mujoco). The trying to use Omniglot dataset not to classify alphabet but to classify characters is nice because the alphabet classification performance is already over 95%, so it is hard to show the gain from your method. However, if the authors validated their method on more complex tasks, it would be helpful to understand or agree their suggestion. # To authers: Thank you for updating the experiment parts, I'd like to update my score by considering it. If the description regarding with comparing with the method using task-average loss or performance on easier tasks is added, it will be more concrete.

Correctness: Correct

Clarity: Well written

Relation to Prior Work: Yes

Reproducibility: Yes

Additional Feedback: - When I read section 1 and 2, I assumed that you evaluate your method with the method having the loss as the sum of the task average loss. I think this method also can solve the underfitting problem on the sparse tasks. So by comparing with it, you can deeply analyze your method I think. - For the case showing worse performance than MAML, analyzing more deeply can be helpful to understand or validate your method. Currently, it is mentioned like TR-MAML showed worse because it more focus on the worst task. - In the aspect that your idea can learn correctly on biased task distribution, it can be related with probabilistic meta-learning methods. As baselines, you can use those methods I think. - It is more fundamental comments, your method learns the tasks beyond given task distribution. However, meta-learning is to learn the task distribution or the shared inductive bias on given tasks. It means your method is to learn more general inductive bias than naive meta-learning method. I think that it can make better performance on out-of-distribution cases. If you show the analysis on those cases, it would be better paper I think.


Review 3

Summary and Contributions: The paper proposes to optimize worst case (meta) loss for MAML to obtain robustness with respect to worst-case test distributions.

Strengths: The work seems to be technically sound and well-executed. The idea makes a lot of sense. The analysis appears to be correct (though I did not check the proofs), and the theoretical results, though simple, are non-trivial and support the claims of the paper. The experiments are performed on relatively simple domains, but the results seem promising. I believe this work will make an impact, and is likely to lead to follow-up work.

Weaknesses: There are two weaknesses, which I think are not critical, but I would appreciate a response from the authors about #1: 1. The mean performance on Omniglot exceeds MAML. But MAML trains for the mean case. Is this not strange? Does it indicate (meta) overfitting on the meta-training set? If so, it would be good to add results for a domain where there are sufficient meta-training examples to avoid overfitting. 2. Following up on #1, if overfitting is the issue, I would recommend a comparison to a regularized variant of MAML (see, e.g. "Meta-Learning without Memorization" Yin et al.). 3. I like the idea behind the paper. But it is a bit obvious -- a less charitable interpretation is that it is a fairly obvious application of known ideas in minimax/DRO to the meta-learning setting. This is the main thing preventing me from giving the paper a higher score. I think in the balance this is OK -- the idea is valuable, and although it is somewhat obvious, there is no previous paper that proposes this, and although the paper utilizes in some sense the most obvious algorithm for solving this problem, it is executed well, and analyzed thoroughly. Other than the two nitpicks above, it's hard to imagine how the authors could have executed on this idea better. If #1 and/or #2 are addressed well in the rebuttal, I would be willing to raise my score to 8. Though perhaps that is not needed to get the paper over the bar.

Correctness: Yes, the claims are correct.

Clarity: Yes, the paper is clearly written, and it was very easy to read.

Relation to Prior Work: Yes, I believe the relevant prior work is generally discussed well. I'm sure there are more citations that could be added (as always...) but I didn't notice glaring omissions.

Reproducibility: Yes

Additional Feedback:


Review 4

Summary and Contributions: This paper proposes TR-MAML, which is a MAML's variant that optimizes for the worst-case performance in the few-shot learning problem. TR-MAML works by replacing the average task loss by the maximum, which is later approximated by a min-max problem over a probability simplex. The authors provide convergence guarantee as well as the generalization bound for TR-MAML. The experiment results support the theoretical claims that TR-MAML can achieve better worst-case performance compared to MAML.

Strengths: The worst-case guarantee in few-shot learning is certainly important. The authors did a good job of providing a clear objective for this problem as well as theoretical analysis for their method. The improvements to Omniglot seem significant.

Weaknesses: My major concern is the insufficient experimental analysis and more concrete experiments are needed. Experimental results on miniImagenet, tieredImagenet dataset would be more informative. Besides, since the authors claim that the proposed method is robust to shifts in the task distribution between meta-training and meta-testing, the experimental result on meta-dataset [1] is more convincing. [1] Triantafillou E, Zhu T, Dumoulin V, Lamblin P, Evci U, Xu K, Goroshin R, Gelada C, Swersky KJ, Manzagol PA, Larochelle H. Meta-Dataset: A Dataset of Datasets for Learning to Learn from Few Examples.

Correctness: The paper seems to be correct.

Clarity: The paper is well written.

Relation to Prior Work: The related work is adequately discussed.

Reproducibility: Yes

Additional Feedback:

[Author Response · NeurIPS 2020]

@R1 **Q: Main contributions.** *Setting:* The idea to minimize the max loss is a novel contribution to meta-learning, as are our notions of task and task instance, needed for the min-max formulation to make sense. *Algorithm:* Our algorithm is an application of the Robust Stochastic Mirror-Prox algorithm [17], but the gradient computations are nontrivial because the task instances must be sampled in such a way that allows for unbiased gradients. *Convergence Analysis:* In the nonconvex setting, we show that our proposed method reaches an $(\epsilon, \delta)$-stationary point (formally defined in (9)) at a rate of $\mathcal{O}(\epsilon^{-5}, \delta^{-5})$ stochastic gradient evaluations, a novel result for the considered stochastic min-max setting. This result is achieved by utilizing novel techniques for evaluating unbiased stochastic gradients of the meta-objective (3) and characterizing their variance, e.g., to prove Theorem 3 we choose a batch size dependent on the number of iterations to show diminishing variance. *Generalization bounds:* The generalization bounds use standard results, but are the first to show convex hull generalization in meta-learning. *Experiments:* Our experiments are the first to evaluate worst-case task performance and show improvements in this metric over a canonical meta-learning algorithm. They also show that TR-MAML can generalize better to new tasks than MAML when the meta-training task distribution is skewed.

**Q: Arguments for minimizing the max-loss not fully convincing.** In many cases, a model that performs well across all tasks is desired, even those considered outlying or "too hard", as well as rare tasks. MAML tends to perform poorly on these tasks since it focuses on average instead of worst-case performance, whereas TR-MAML prioritizes performance on them. TR-MAML may also generalize better since it focuses on a wider range of meta-training tasks.

**Q: Generalization bounds.** The convex hull generalization result highlights the advantage of minimizing the max loss in the sense that a wider range of tasks are prioritized during meta-training, and does not hold for MAML. Assuming nonnegative and $M$-smooth $f_i$ and $\alpha < 1/M$, we have $0 \leq f_i(w - \alpha \nabla f_i(w)) \leq f_i(w)$, so ignoring gradient noise, the Rademacher complexity of the function after one step of gradient descent is at most the standard Rademacher complexity, which is well-known for many classes of functions. We will include this as a corollary in the revised paper.

@R2 **Q: Need to make the task.** Indeed, we must construct the tasks such that the min-max problem over them is tractable. This construction is natural in many settings, for example in the Omniglot experiment, where we aim to meta-learn how to classify characters from the same alphabet. For continuous distributions of task instances, it is reasonable to expect that the discretization into tasks would also be naturally suggested by the setting.

**Q: Case showing worse performance than MAML, analyze more deeply.** Assuming reference to the $(5, 1)$ Omniglot case, our claim is that minimizing the max loss leads to better average generalization in *some* cases, especially those with very different average meta-train and meta-test task instances. We suspect this is not true for the $(5, 1)$ case.

**Q: Better performance on OOD cases.** We provably show generalization within the convex hull of meta-training tasks (Theorem 4), but do not believe it is possible to provably show generalization beyond this.

@R4 **Q: Mean performance on Omniglot exceeds MAML**. *Regarding meta-train performance* (also concerns @R2 **compare to method with task average loss**): 'Weighted Mean' is the uniform average over task instances (i.e. is the surrogate for the expected loss over tasks given in Equation 1), and weighs the average accuracy on each task (alphabet) by the number of instances it contains. MAML aims to minimize this metric, and always outperforms TR-MAML on it. 'Mean' is the uniform average accuracy across tasks. These two statistics show that TR-MAML treats the tasks more uniformly than MAML, performing worse on the most frequent tasks (yielding a smaller 'Weighted Mean') but better on the rare tasks (larger 'Mean'). *Regarding meta-test performance*: these results support our claim that TR-MAML can generalize better than MAML because it prioritizes performance on all the meta-training tasks. We include an experiment below on Mini-ImageNet in which the tasks contain a broader range of images (a random selection of image classes, instead of only characters from the same alphabet, as well as many more images per class), so we expect that MAML overfitting to the most popular tasks is less likely, but similar relative performance is observed.

@R1 R2 R5 **Q: More experiments.** Thanks for the feedback. In the revision, we'll include experiments on Mini-ImageNet. We split the image classes into two subsets: 64 classes used for meta-training, and the remaining 36 for meta-testing. We create tasks as follows: we randomly group the 64 meta-training classes into 8 meta-train tasks, with the numbers of classes/task being $\{6, 7, 7, 8, 8, 9, 9, 10\}$. Likewise, the 36 meta-test classes are randomly split into 4 tasks, each with 9 classes/task. Each task instance is constructed by sampling 1 image each from 5 distinct classes within a task: thus, this is 5-way 1-shot problem. We meta-train for 60k iterations with a batch size of 2 task instances, and 5 steps of gradient descent for local adaptation. Our results show the Weighted Mean accuracy (aka average case over task instances) and the worst-case performance (aka worst accuracy over the tasks). The first two columns are generated by testing on *new task instances* from the meta-training classes; the second two columns are generated by testing on task instances from the previously unseen meta-test classes. We give 95% confidence intervals over 3 trials.

Table 1: Mini-ImageNet 5-way, 1-shot accuracies

| $(N, K)$ | Algorithm | Eight Meta-Training Tasks | | Four Meta-Testing Tasks | |
|---|---|---|---|---|---|
| | | Weighted Mean | Worst | Weighted Mean | Worst |
| (5,1) | MAML | $\mathbf{70.1 \pm 2.2}$ | $48.0 \pm 4.5$ | $46.6 \pm .4$ | $44.7 \pm .7$ |
| | TR-MAML | $63.2 \pm 1.3$ | $\mathbf{60.7 \pm 1.6}$ | $\mathbf{48.5 \pm .6}$ | $\mathbf{45.9 \pm .8}$ |

[Meta-Review · NeurIPS 2020]

This paper stirred a lot of discussion between reviewers. The reviewers appreciated the new experiments on MiniImagenet. The primary outstanding reviewer concerns were: (a) limited motivation for using a max loss (particularly since, if tasks are of varying difficulty, then the max loss will focus solely on the hardest task rather than taking all tasks into account) (b) the finding that the proposed method is doing better on average case loss than standard MAML despite optimizing worst-case loss. (c) the discrepancy between the standard meta-learning experimental set-up and the one in the paper + author response (e.g. with 8 training tasks and 4 test tasks) I think that none of these reasons are grounds for rejecting the paper. However, I strongly encourage the authors to revise the paper for the camera ready to address these concerns. In particular, to address (a), the authors are encouraged to revise the paper to discuss settings where the max loss may not have the desired effect. To address (b), the authors are encouraged to further analyze this phenomenon and discuss it in the paper. To address (c), the authors are encouraged to include results on the standard set-up in the supplemental material.